# CAN WE EVALUATE DOMAIN ADAPTATION MODELS WITHOUT TARGET-DOMAIN LABELS?

**Jianfei Yang**[1]*, **Hanjie Qian**[1]*, **Yuecong Xu**[2], **Kai Wang**[2], **Lihua Xie**[1].
[1]Nanyang Technological University    [2]National University of Singapore

## ABSTRACT

Unsupervised domain adaptation (UDA) involves adapting a model trained on a label-rich source domain to an unlabeled target domain. However, in real-world scenarios, the absence of target-domain labels makes it challenging to evaluate the performance of UDA models. Furthermore, prevailing UDA methods relying on adversarial training and self-training could lead to model degeneration and negative transfer, further exacerbating the evaluation problem. In this paper, we propose a novel metric called the *Transfer Score* to address these issues. The proposed metric enables the unsupervised evaluation of UDA models by assessing the spatial uniformity of the classifier via model parameters, as well as the transferability and discriminability of deep representations. Based on the metric, we achieve three novel objectives without target-domain labels: (1) selecting the best UDA method from a range of available options, (2) optimizing hyperparameters of UDA models to prevent model degeneration, and (3) identifying which checkpoint of UDA model performs optimally. Our work bridges the gap between data-level UDA research and practical UDA scenarios, enabling a realistic assessment of UDA model performance. We validate the effectiveness of our metric through extensive empirical studies on UDA datasets of different scales and imbalanced distributions. The results demonstrate that our metric robustly achieves the aforementioned three goals.[1]

## 1 INTRODUCTION

Deep neural networks have made significant progress in a wide range of machine learning tasks. However, training deep models typically requires large amounts of labeled data, which can be costly or difficult to obtain in some cases. To overcome this challenge, unsupervised domain adaptation (UDA) has emerged as a technique for transferring knowledge from a labeled source domain to an unlabeled target domain. For example, in autonomous driving, UDA enables a deep segmentation model trained on data from normal weather conditions to adapt to rainy, hazy, and snowy weather conditions where the data distribution changes dramatically (Liu et al., 2020).

Despite the improvement of performance realized by UDA models, the current evaluation process relies on target-domain labels that are usually unavailable in real-world scenarios. As a result, it can be difficult to determine the effectiveness of UDA methods and how well they perform in the target domain. Moreover, since adversarial training is widely used in domain adaptation (Wang & Deng, 2018), many UDA models can be prone to unstable training processes and even negative transfer if the hyperparameters are not well selected (Wang et al., 2019a), which can further undermine the viability of UDA models in practice. Therefore, there is a pressing need to develop an unsupervised evaluation method for UDA models.

To evaluate UDA models in an unsupervised manner, we contemplate whether the domain discrepancy metric could be a good indicator of model performance. The maximum mean discrepancy (MMD) is commonly used to measure the statistical difference in the Hilbert space between the source and target domains (Gretton et al., 2012). Proxy A-distance (PAD) is established on the statistical learning theory of UDA (Ben-David et al., 2010) and measures distribution discrepancy by training a binary

---

*Equal contribution (jianfei.yang@ntu.edu.sg, hanjie001@ntu.edu.sg).
[1]Codes are available at https://sleepyseal.github.io/TransferScoreWeb/

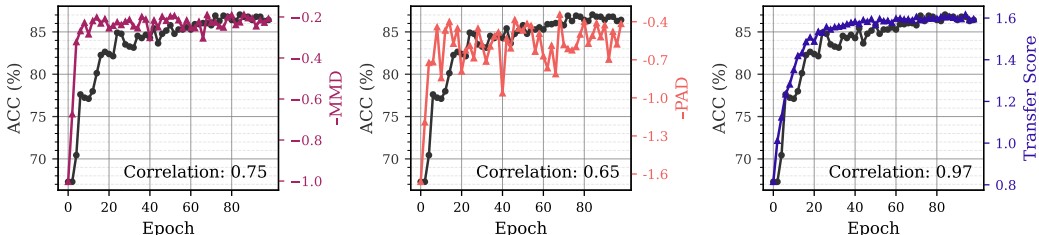

Figure 1: A preliminary experiment of a classic UDA method (DANN (Guan & Liu, 2021)) on Office-31 A→W task. The correlation between existing domain discrepancy metrics (MMD and Proxy A-distance) and the target-domain accuracy is not clear while our proposed *Transfer Score* could clearly indicate the target-domain accuracy with a high correlation value (Pearson, 1895).

classifier on the two domains (Ben-David et al., 2010). However, as shown in Fig. 1, our preliminary results suggest that these metrics fail to provide an accurate evaluation of UDA model performance in the target domain, and they cannot indicate the negative transfer phenomenon (Wang et al., 2019a).

In this paper, we propose an unsupervised metric for UDA model evaluation and selection, addressing three crucial challenges in real-world UDA scenarios: (1) selecting the best UDA model from a range of UDA method candidates; (2) adjusting hyperparameters to prevent the negative transfer and achieve enhanced performance; and (3) identifying the optimal checkpoint (epoch) of model parameters to avoid overfitting. To this end, the *Transfer Score* is proposed to accurately indicate the UDA model's performance in the target domain. The TS metric evaluates UDA models from two perspectives. First, it evaluates the spatial uniformity of the model directly from model parameters to determine whether the classifier is biased or overfitted for all classes. Second, it evaluates the transferability and discriminability of deep representations after UDA by calculating the clustering tendency and the mutual information. We conduct extensive experiments on public datasets (Office-31, Office-Home, VisDA-17, and DomainNet) using five categories of UDA methods, and our results demonstrate that the proposed metric effectively resolves the aforementioned challenges. As far as we know, TS is the first work to study and achieve *simultaneous unsupervised UDA model comparison and selection*.

## 2 RELATED WORK

### 2.1 UNSUPERVISED DOMAIN ADAPTATION

Current UDA methods aim to extract common features across labeled source domains and unlabeled target domains, improving the transferability of representations and robustness of models while mitigating the burden of manual labeling (Long et al., 2016). Most UDA methods could be generally divided into two categories: i) adversarial-based methods (Ganin & Lempitsky, 2015; Jiang et al., 2020; Xu et al., 2021; Yang et al., 2020a; 2021b), where domain-invariant features are extracted by an adversarial training where a domain discriminator is trained against feature extractor (Huang et al., 2011; Ganin & Lempitsky, 2015); and ii) metric-based methods (Zhang et al., 2019; Yang et al., 2021a; Sun et al., 2016), which mitigate domain shifts across domains by applying metric learning approaches, minimizing metrics such as MMD (Gretton et al., 2012; Long et al., 2015; 2016; Xu et al., 2022), and PAD (Han et al., 2022; Xie et al., 2022). So far, there have emerged various streams of UDA approaches including reconstruction-based methods (Yang et al., 2020b; Yeh et al., 2021; Li et al., 2022), norm-based methods (Xu et al., 2019), and reweighing-based methods (Jin et al., 2020; Wang et al., 2022; Xu et al., 2023).

### 2.2 MODEL EVALUATION AND VALIDATION OF UDA

The current evaluation process of UDA methods is not feasible in real-world applications since it relies on target-domain labels that are unavailable in real-world scenarios. There exists various domain discrepancy metrics such as MMD (Gretton et al., 2012) and PAD (Ben-David et al., 2010) that measure the discrepancies between source and target distribution. However, they only represent the cross-domain distribution shift under a certain feature space that cannot be directly related to model performance and used for model evaluation and selection.

Few researchers pay attention to UDA model evaluation without target-domain annotations. In particular, Stable Transfer (Agostinelli et al., 2022) aims to analyze the different transferability metrics (Tran et al., 2019; Nguyen et al., 2020; You et al., 2021; 2022; Ma et al., 2021) for model selection in transfer learning under a pre-training and fine-tuning mechanism where both the source and target datasets are similar and labeled. Yet, the analysis does not apply to the task of UDA where there is a significant domain shift while the target domain is unlabeled. Unsupervised validation methods aim to choose a better model by cross-validation or hyperparameter tuning. DEV (You et al., 2019) firstly estimates the target risk based on labeled validation sets and massive iterations, which still assumes the viability of partial target-domain labels and takes up huge computation costs. There are other methods focusing on unsupervised validation including entropy-based method (Morerio et al., 2017), geometry-based method (Saito et al., 2021), and out-of-distribution method (Garg et al., 2022). They only focus on model selection but have not explored how to choose a better UDA model from various candidates. Whereas, our transfer score can perform UDA method comparison and model selection simultaneously without the cumbersome iterative process.

## 3 PRELIMINARY

### 3.1 UNSUPERVISED DOMAIN ADAPTATION

In unsupervised domain adaptation, we assume that a model is learned from a labeled source domain $\mathcal{D}_S = \{(x_i^s, y_i^s)\}_{i \in [N_s]}$ and an unlabeled target domain $\mathcal{D}_T = \{x_i^t\}_{i \in [N_t]}$. The label space $Y$ is a finite set ($Y = 1, 2, ..., K$) shared between both domains. Assume that the source domain and the target domain have different data distributions. In other words, there exists a domain shift (*i.e.*, covariate shift) (Ben-David et al., 2010) between $\mathcal{D}_S$ and $\mathcal{D}_T$. The objective of UDA is to learn a model $\Phi(\cdot) = h \circ g$ where $h$ denotes the classifier and $g$ denotes the feature extractor, which can predict the label $y_i^t$ given the target-domain input $x_i^t$.

### 3.2 CHALLENGE: CAN WE EVALUATE UDA MODELS WITHOUT TARGET-DOMAIN LABELS?

Despite the significant progress made in UDA approaches (Wang & Deng, 2018), most existing methods still require access to target-domain labels for model selection and evaluation. This limitation poses a challenge in real-world scenarios where obtaining target-domain labels is often infeasible. This issue hinders the practical applicability of UDA methods. Moreover, current UDA approaches heavily rely on adversarial training (Ganin et al., 2016) and self-training techniques (*i.e.*, pseudo labels) (Kumar et al., 2020; Cao et al., 2023) that often yield unstable adaptation outcomes, further impeding the effectiveness of UDA in practical settings. Thus, there is a pressing need to develop methods that enable unsupervised model evaluation in UDA without relying on target-domain labels.

To address this issue, one may leverage metrics that measure the distribution discrepancy between the source and target domains to indicate model performance, as these metrics represent feature transferability (Pan et al., 2011). We consider two common metrics for UDA evaluation: maximum mean discrepancy (MMD) (Gretton et al., 2007) and proxy A-distance (PAD) (Ben-David et al., 2010). MMD is a statistical test that determines whether two distributions $p$ and $q$ are the same. MMD is estimated by $\text{MMD}^2(p, q) = \|\mathbb{E}_p[\phi(X_S)] - \mathbb{E}_q[\phi(X_T)]\|_{\mathcal{H}_k}^2$ where $\phi(\cdot)$ maps the input to another space and $\mathcal{H}_k$ denotes the Reproducing Kernel Hilbert Space (RKHS). PAD is a measure of domain disparity between two domains established by the statistical learning theory (Ben-David et al., 2010). PAD is defined as $d_A = 2(1 - 2\epsilon)$ where $\epsilon$ is the error of a domain classifier, *e.g.*, SVM. We performed a preliminary experiment using MMD and PAD to indicate the model's performance. As depicted in Fig. 1, we observed a negative correlation between the target-domain accuracy and both MMD (correlation value of 0.75) and PAD (correlation value of 0.65). However, neither of these metrics provides a clear indication of the target-domain accuracy that could help model selection. Consequently, they are insufficient for evaluating and selecting UDA models in real-world scenarios.

This paper introduces a novel metric called the *Transfer Score*, which has a strong correlation with target-domain accuracy. It serves three primary objectives in real-world UDA scenarios: (1) selecting the most appropriate UDA method from a range of available options, (2) optimizing hyperparameters to achieve enhanced performance, and (3) identifying the epoch at which the adapted model performs optimally. As illustrated in Fig. 1, our proposed metric exhibits a significantly higher correlation value of 0.97 with target-domain accuracy, showcasing its efficacy in real-world UDA scenarios.

## 4 AN UNSUPERVISED METRIC: TRANSFER SCORE

As an unsupervised metric for UDA evaluation, the Transfer Score (TS) relies on the evaluation of model parameters and feature representations, which are readily available in real-world UDA scenarios.

### 4.1 MEASURING UNIFORMITY FOR CLASSIFIER BIAS

Model parameters encapsulate the intrinsic characteristics of a deep model, as they are independent of the data. However, understanding the feature space solely based on these parameters is challenging, especially for complex feature extractors such as CNN (LeCun et al., 1998) and Transformer (Vaswani et al., 2017). Therefore, we defer the evaluation of the feature space to a data-driven metric, discussed in Section 4.2. In this section, we focus on the transferability of the classifier via model parameters. In cross-domain scenarios, we observe that a classifier trained on the source domain often exhibits biased predictions when applied to the target domain. This bias stems from an over-emphasis on certain classes in the source domain due to their larger number of samples (Jamal et al., 2020). Consequently, we hypothesize that a classifier with superior transferability should divide the feature space evenly and generate class-balanced prediction, rather than disproportionately emphasizing specific classes. Prior theoretical research has also demonstrated that evenly partitioning the feature space leads to improved model generalization ability (Wang et al., 2020).

Now, let's consider how to measure the uniformity of the feature space divided by a classifier. We propose that the uniformity is reflected by the consistency of the angles between the decision hyperplanes of the classifier. Let $h(\cdot) \in \mathbb{R}^{d \times K}$ denote a $K$-way classifier comprising $K$ vectors $[w_1, w_2, ..., w_K]$, which maps a $d$-dimensional feature to a prediction vector. When the feature space is evenly partitioned by the classifier, the angles between any two vectors among the $K$ vectors of the classifier are equal. We denote the ideal angle as $\theta_K$ and define the angle matrix of the classifier as $\Sigma_h \in \mathbb{R}^{K \times K}$, where each entry $\theta_{ij}$ is the angle between $w_i$ and $w_j$. The uniformly distributed angle matrix is defined as $\Sigma_u \in \mathbb{R}^{K \times K}$, where each entry corresponds to the ideal angle $\theta_K$. Notably, the diagonal entries of both $\Sigma_h$ and $\Sigma_u$ are all set to 0.

**Definition 1.** *The uniformity of $h(\cdot)$ is the square of the Frobenius norm of the difference matrix between $\Sigma_h$ and $\Sigma_u$:*

$$\mathcal{U} = \frac{1}{2}\|\Sigma_h - \Sigma_u\|_F^2 = \frac{1}{K(K-1)} \sum_{i=1}^{K} \sum_{j=1}^{K} (\theta_{ij} - \theta_K)^2 \,, \tag{1}$$

*where $\| \cdot \|_F$ is the Frobenius norm of a matrix.*

Intuitively, this metric can be interpreted as the mean squared error between all the cross-hyperplane angles and the ideal angle. The smaller value indicates a more transferable classifier. We further provide a closed-form formula for computing the ideal angle $\theta_K$, which is proven in the *Appendix*.

**Theorem 1.** *When $K \leq d + 1$, the ideal angle $\theta_K$ can be calculated by*

$$\theta_K = arccos\left(-\frac{1}{K-1}\right). \tag{2}$$

In practical UDA applications, the feature dimension $d$ is typically greater than the number of classes (Saenko et al., 2010; Venkateswara et al., 2017; Peng et al., 2019). Consequently, the uniformity metric $\mathcal{U}$ can be easily computed with Eq.(1).

### 4.2 MEASURING FEATURE TRANSFERABILITY AND DISCRIMINABILITY

In addition to evaluating the transferability of the classifier, we also assess the transferability and discriminability of the feature space, as they directly reflect the target-domain performance of UDA (Chen et al., 2019). As evaluating the feature space based on model parameters is challenging, we propose to resort to data-driven metrics: Hopkins statistic and mutual information.

Firstly, we propose to leverage the Hopkins statistic (Banerjee & Dave, 2004) as a metric to measure the clustering tendency of the feature representation in the target domain. The Hopkins statistic,

belonging to the family of sparse sampling tests, assesses whether the data is uniformly and randomly distributed and measures the clarity of the clusters. For a good UDA model, the feature space should exhibit distinct clusters for each class, indicating better transferability and discriminability (Deng et al., 2019; Li et al., 2021). Conversely, if the samples are randomly and uniformly distributed, achieving high classification accuracy becomes challenging. Therefore, we assume that the target-domain accuracy should be correlated with the Hopkins statistic. To compute the Hopkins statistic, we start by generating a random sample set $R$ comprising $m \ll N_t$ data points, sampled without replacement, from the feature embeddings of the target domain samples $g(x)$. Additionally, we generate a set $U$ of $m$ uniformly and randomly distributed data points. Next, we define two distance measures: $u_i$, which represents the distance of samples in $U$ from their nearest neighbors in $R$, and $w_i$, which represents the distance of samples in $R$ from their nearest neighbors in $R$. The Hopkins statistic is then defined as follows:

$$\mathcal{H} = \frac{\sum_{i=1}^{m} u_i^d}{\sum_{i=1}^{m} u_i^d + \sum_{i=1}^{m} w_i^d}, \tag{3}$$

where $d$ denotes the dimension of the feature space.

The Hopkins statistic evaluates the distribution of samples within the feature space generated by the extractor $g(\cdot)$. However, it does not provide insights into how the classifier $h(\cdot)$ behaves within this feature space. As a result, even in scenarios where all samples form a single cluster or the classifier boundary intersects a densely populated region of samples, the Hopkins statistic can still yield a high value. To address this limitation, we propose the utilization of mutual information between the input and prediction in the target domain. By incorporating mutual information, we can discern the prediction confidence and diversity (class balance) of the UDA model, thereby reflecting the transferability of features in the target domain. The mutual information $\mathcal{M}$ is defined as:

$$\mathcal{M} = H(\mathbb{E}_{x \in \mathcal{D}_t} h(g(x))) - \mathbb{E}_{x \in \mathcal{D}_t} H(h(g(x))), \tag{4}$$

where $H(\cdot)$ denotes the information entropy. As the mutual information value measures how well the model adheres to the cluster assumption, it serves as a regularizer for domain adaptation in various works such as DIRT-T (Shu et al., 2018), DINE (Liang et al., 2022), BETA (Yang et al., 2022a), and semi-supervised learning approaches (Grandvalet & Bengio, 2005).

## 4.3 TRANSFER SCORE

Consolidating the uniformity $\mathcal{U}$ which evaluates the classifier $h(\cdot)$ and the feature transferability metrics $\mathcal{H}, \mathcal{M}$ which assess the feature space generated by the feature extractor $g(\cdot)$, we introduce the Transfer Score to evaluate the target-domain model $\Phi(\cdot) = h \circ g$:

**Definition 2.** *The Transfer Score is given by*

$$\mathcal{T} = -\mathcal{U} + \mathcal{H} + \frac{|\mathcal{M}|}{\ln K}. \tag{5}$$

*where $K$ is the number of classes for the normalization purpose.*

We aim to use a larger transfer score to indicate better transferability. To this end, as the greater uniformity indicates a larger bias and lower transferability, we use the negative uniformity. In contrast, we use the Hopkins statistic and the absolute value of mutual information as they are positively correlated to the transferability. The mutual information is especially normalized because it does not have a fixed range as the uniformity and Hopkins statistic does. Our TS serves two purposes for UDA: (1) comparing different UDA methods to select the most suitable one, and (2) assisting in hyperparameter tuning.

**Saturation Level of UDA Training.** It has been observed that UDA does not consistently lead to improvement for deep models (Wang et al., 2019b). Due to potential overfitting, the highest target-domain accuracy is often achieved during the training process, but the current UDA methods directly use the last-epoch model. To determine the optimal epoch for model selection after UDA, we introduce the saturation level of UDA training, represented by the coefficient of variation of the TS.

**Definition 3.** *Denote $\mathcal{T}_m$ as the Transfer Score at epoch $m$. The saturation level is defined as*

$$\mathcal{S}_m = \frac{\sigma_m}{\mu_m}, \tag{6}$$

*where $\sigma_m$ and $\mu$ are the standard deviation and mean within a sliding window $\tau$, respectively.*

When the saturation level of the TS falls below a predefined threshold $\zeta$, it indicates that the TS has reached a point of saturation. Beyond this threshold, training the model further could potentially result in a decline in performance. Therefore, to determine the optimal checkpoint, we select the epoch with the highest TS within that time window. This approach does not necessarily select the best-performing model but effectively enables us to mitigate the risk of performance degradation caused by continued training and overfitting.

## 5  EMPIRICAL STUDIES

### 5.1  SETUP

**Dataset.** We employ four datasets in our studies for different purposes. **Office-31** (Saenko et al., 2010) is the most common benchmark for UDA including three domains (**A**mazon, **W**ebcam, **D**SLR) in 31 categories. **Office-Home** (Venkateswara et al., 2017) is composed of four domains (**Ar**t, **Cl**ipart, **Pr**oduct, **Re**al World) in 65 categories with distant domain shifts. **VisDA-17** (Peng et al., 2017) is a synthetic-to-real object recognition dataset including a source domain with 152k synthetic images and a target domain with 55k real images from Microsoft COCO. **DomainNet** (Peng et al., 2019) is the largest DA dataset containing 345 classes in 6 domains. We adopt two imbalanced domains, Clipart (c) and Painting (p).

**Baseline.** To thoroughly evaluate the effectiveness and robustness of our metric across different UDA methods, we select classic UDA baseline methods including five types: adversarial UDA method (DANN (Ganin et al., 2016), CDAN (Long et al., 2018), MDD (Zhang et al., 2019)), moment matching method (DAN (Long et al., 2017), CAN (Kang et al., 2019)), norm-based method (SAFN (Xu et al., 2019)), self-training method (FixMatch (Sohn et al., 2020), SHOT (Liang et al., 2021), CST (Liu et al., 2021), AaD (Yang et al., 2022b)) and reweighing-based method (MCC (Jin et al., 2020)). For comparison, we choose recent works on unsupervised validation of UDA and out-of-distribution (OOD) model evaluation methods as our comparative baselines: C-Entropy (Morerio et al., 2018) based on entropy, SND (Saito et al., 2021) based on neighborhood structure, ATC (Garg et al., 2022) for OOD evaluation, and DEV (You et al., 2019).

**Implementation Details.** For the Office-31 and Office-Home datasets, we employ ResNet-50, while ResNet-101 is used for VisDA-17 and DomainNet. The hyperparameters, training epochs, learning rates, and optimizers are set according to the default configurations provided in their original papers. We set the hyperparameters $\tau = 3$ and $\zeta = 0.01$ based on simple validation conducted on the Office-31 dataset, which performs well across all other datasets. Each experiment is repeated three times, and the reported results are the mean values with standard deviations. All the figures report the mean results except Fig. 4 which visualizes specific training procedures. We have included a detailed implementation of empirical studies in the supplementary materials.

### 5.2  TASK 1: SELECTING A BETTER UDA METHOD

We assess the capability of TS in comparing and selecting UDA methods. To this end, we train UDA baseline models on Office-Home and calculate the TS at the last training epoch. The results of TS are visualized with the target-domain accuracy in Fig. 2. It is shown that the highest TS accurately indicates the best UDA method for four tasks, and the TS even reflects the tendency of

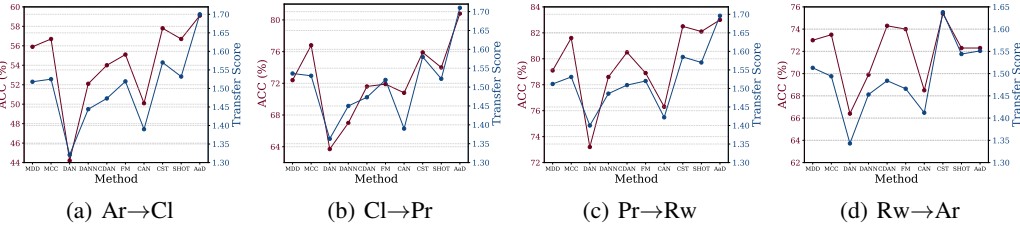

|  (a) Ar→Cl | (b) Cl→Pr | (c) Pr→Rw | (d) Rw→Ar |

Figure 2: The relationship of accuracy and TS among different methods on four tasks of Office-Home. The proposed TS can accurately indicate the target-domain performance. (FM denotes FixMatch.)

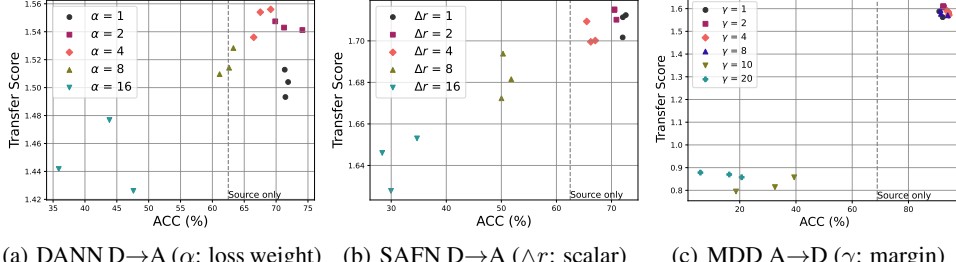

(a) DANN D→A ($\alpha$: loss weight)   (b) SAFN D→A ($\triangle r$: scalar)   (c) MDD A→D ($\gamma$: margin)

Figure 3: The TS and the target-domain accuracy under different hyperparameter settings. The models with better hyperparameters are reflected by higher TS.

the target-domain accuracy across different UDA methods. Thus, our metric proves to be effective in selecting a good UDA method from various UDA candidates, but it may not necessarily choose the absolute best-performing model if the performance difference is very small. For Task 1, though existing unsupervised validation methods (e.g., SND (Saito et al., 2021) and C-Entropy (Morerio et al., 2018)) do not consider UDA model comparison, their scores can be tested for Task 1. The results are discussed in the appendix, which shows that existing approaches cannot achieve Task 1.

### 5.3    TASK 2: HYPERPARAMETER TUNING FOR UDA

After selecting a UDA model, it is crucial to perform hyperparameter tuning as inappropriate hyperparameters can lead to decreased performance and even negative transfer (Wang et al., 2019b). We evaluate TS via three methods with their hyperparameters: DANN with the weight of adversarial loss $\alpha$, SAFN with the residual scalar of feature-norm enlargement $\triangle r$, and MDD with the margin $\gamma$. Their target-domain accuracies and TS on Office-31 are shown in Fig. 3, where models with higher TS show significant improvement after adaptation. it is observed that unsuitable hyperparameters can result in performance degradation, sometimes even worse than the source-only model. Overall, the TS metric proves to be valuable in guiding hyperparameter tuning, allowing us to avoid unfavorable outcomes caused by inappropriate hyperparameter choices.

### 5.4    TASK 3: CHOOSING A GOOD CHECKPOINT AFTER TRAINING

The selection of an appropriate checkpoint (epoch) is crucial in UDA, as UDA models often tend to overfit the target domain during training. In Fig. 4, it is observed that all the baseline UDA methods experience a decline in performance after epochs of training. Notably, SAFN on DomainNet (c→p) exhibits a significant drop in accuracy of over 17.0%. We utilize the saturation level of TS to identify a good model checkpoint (marked as a star) within the window size (in red). Detailed results on 7 UDA methods are listed in Tab. 1. Compared to the last epoch (i.e., an empirical choice), our method works well on most UDA baseline methods, demonstrating a robust strategy to choose a reliable checkpoint while overcoming the negative transfer due to the overfitting issue.

We compare our method with the recent works on model evaluation for UDA and out-of-distribution tasks in Tab. 2. We find that our method outperforms all other methods. C-Entropy cannot choose a better model checkpoint on many tasks, since the entropy only reflects the prediction confidence, which has been enriched by more perspectives in our method. The SND leverages neighborhood structure for UDA model evaluation, but it has a very high computational complexity. It is noteworthy that all other methods cannot achieve the goal of task 1 and 2.

Table 1: The model accuracy (%) of the last epoch and the epoch chosen by our method.

| Dataset | VisDA-17 | | | DomainNet (c→p) | | | DomainNet (p→c) | | |
|---|---|---|---|---|---|---|---|---|---|
| Method | Last | Ours | Imp. ↑ | Last | Ours | Imp. ↑ | Last | Ours | Imp. ↑ |
| DAN | 66.9±0.4 | 68.3±0.3 | +1.4 | 35.6±0.5 | 36.8±0.3 | +1.2 | 45.6±0.3 | 45.6±0.5 | - |
| DANN | 72.9±0.5 | 73.8±0.3 | +0.9 | 36.0±0.5 | 37.9±0.3 | +1.9 | 34.5±0.3 | 40.2±0.4 | +5.7 |
| AFN | 58.8±0.6 | 74±0.5 | +15.2 | 29.6±5.8 | 41.0±0.5 | +11.4 | 39.1±0.6 | 45.9±0.2 | +6.8 |
| CDAN | 76.2±0.7 | 76.4±0.6 | +0.2 | 39.5±0.2 | 39.8±0.2 | +0.3 | 44.1±0.4 | 44.8±0.3 | +0.7 |
| MDD | 71.4±3.0 | 74.5±1.5 | +3.1 | 42.9±0.2 | 42.5±0.1 | -0.4 | 48.0±0.3 | 48.2±0.4 | +0.2 |
| MCC | 76.4±0.7 | 79.5±0.6 | +3.1 | 32.9±0.8 | 37.3±0.1 | +4.4 | 44.6±0.3 | 44.8±0.4 | +0.2 |
| FixMatch | 49.2±0.9 | 66.6±0.2 | +17.4 | 40.1±0.3 | 41.5±0.2 | +1.4 | 52.7±0.3 | 53.2±0.5 | +0.5 |

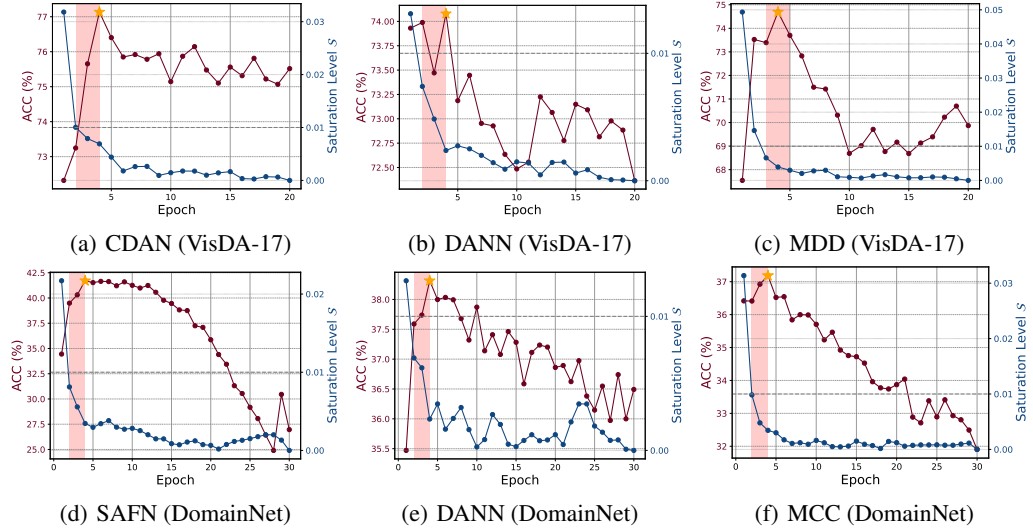

Figure 4: The choice of model checkpoint after UDA training for different methods.

Table 2: Comparison of different methods on Office-31 (W→A), Office-Home (Rw→Ar) and VisDA-17.

| | | CDAN | | | MCC | | |
|---|---|---|---|---|---|---|---|
| Task | Publication | W→A | Rw→Ar | VisDA-17 | W→A | Rw→Ar | VisDA-17 |
| Source-Only | - | 65.5 | 62.3 | 72.6 | 70.2 | 65.6 | 71.7 |
| DEV | ICML-19 | 66.4 | 63.5 | 72.6 | 67.6 | 63.1 | 72.3 |
| C-Entropy | ICLR-18 | 63.8 | 61.7 | 69.9 | 72.4 | 66.9 | 68.9 |
| SND | ICCV-21 | 67.0 | 70.8 | 70.3 | 67.4 | 68.8 | 73.0 |
| ATC | ICLR-22 | 70.7 | 73.5 | 75.1 | 73.9 | 73.0 | 78.1 |
| Ours | - | **73.3** | **74.0** | **76.4** | **74.5** | **73.3** | **79.5** |

Table 3: The ablation study on three UDA methods.

| $\mathcal{U}$ | $\mathcal{H}$ | $\mathcal{M}$ | MCC | CST | AaD |
|---|---|---|---|---|---|
| ✓ | | | 35.8 | 81.8 | 84.9 |
| | ✓ | | 35.8 | 82.2 | 84.9 |
| | | ✓ | 35.9 | 82.7 | 84.9 |
| ✓ | ✓ | | 36.0 | 83.8 | 85.3 |
| | ✓ | ✓ | 36.5 | 83.3 | 85.8 |
| ✓ | | ✓ | 36.5 | 83.3 | 85.6 |
| ✓ | ✓ | ✓ | 37.3 | 83.8 | 85.9 |

## 5.5 ANALYTICS

**Ablation Study of Metrics.** To assess the robustness of each metric in the TS, we performed a checkpoint selection experiment on MCC and DomainNet (c→p) using three independent metrics: uniformity, Hopkins statistic, and mutual information. As summarized in Tab. 3, the uniformity, Hopkins statistic, and mutual information achieve the accuracies of 35.8%, 35.8%, and 35.9%, respectively, slightly lower than the accuracy obtained using TS (37.3%). Two more experiments are provided on VisDA-17 for CST and AaD.

**Evaluation on Imbalanced Dataset.** We explore the viability of our method on a large-scale long-tailed imbalanced dataset, DomainNet. As shown in Fig. 5(a), the label distributions of these two domains are long-tailed and shifted. In Fig. 5(b), it is shown that TS can accurately indicate the performance of various UDA methods, successfully achieving task 1. As illustrated in Tab. 1 and Fig. 4, TS can stably improve UDA methods by choosing a better model checkpoint on DomainNet. We further explore the model uniformity on the imbalanced dataset. In Fig. 6(a) and Fig. 6(b), the increased uniformity reflects decreasing accuracy during training, indicating the model becomes more biased, which explains why these UDA methods perform worse on the imbalanced dataset.

**Hyperparameter Sensitivity.** We study the sensitivity of $\tau$ using DAN and MCC on the DomainNet, varying $\tau$ within the range of $[2, 7]$. The results, depicted in Fig. 6(c), indicate that the best performance is achieved when the window size is set to 3. This finding suggests that considering a relatively short range of TS values has been sufficient for UDA checkpoint selection. Our method also includes another hyper-parameter $\zeta$. $\zeta$ controls the variation of accuracy within the sliding window $\tau$ when the saturation point is chosen. We conducted experiments on all four datasets and found that when the UDA models converge, such variations always do not exceed 1.0%. Thus, we just need to set $\zeta$ to be sufficiently small, *i.e.*, 0.01.

**t-SNE Visualization and Uniformity of Classifier.** To gain a more intuitive understanding of TS, we visualize target-domain features using t-SNE (Van der Maaten & Hinton, 2008) and the classifier parameters using a unit circle in Fig. 7 where each radius line in the unit circle corresponds to a

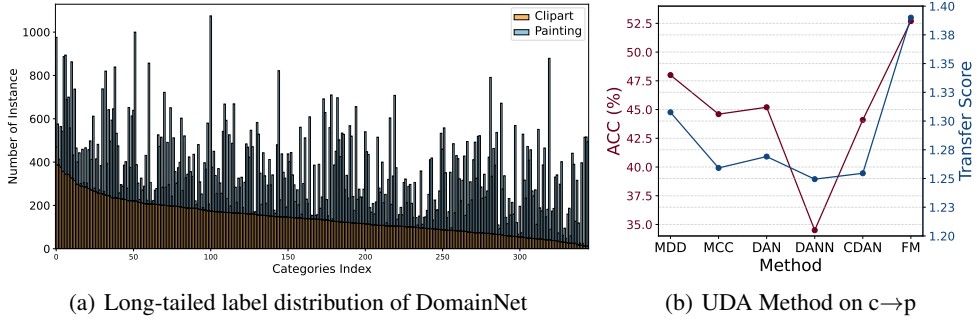

(a) Long-tailed label distribution of DomainNet

(b) UDA Method on c→p

Figure 5: Empirical studies on the imbalanced dataset DomainNet.

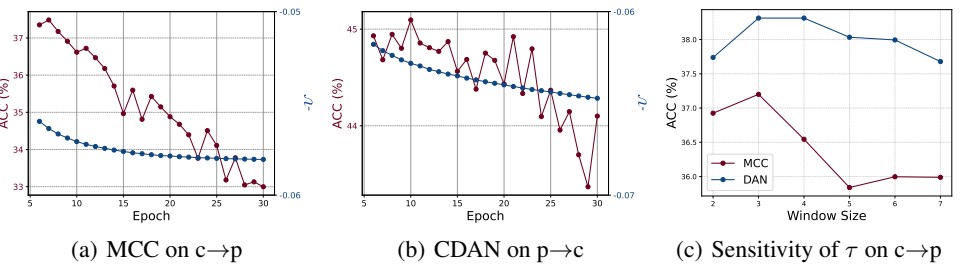

(a) MCC on c→p

(b) CDAN on p→c

(c) Sensitivity of $\tau$ on c→p

Figure 6: The uniformity and sensitivity study on the imbalanced dataset DomainNet.

classifier vector $w_i$ after dimension reduction. The visualization includes the source-only model (S.O.), DAN, and MCC, with the accuracy ranking of S.O.<DAN<MCC. It is found that a higher Hopkins statistic value accurately captures the better clustering tendency, and the classifiers with better uniformity perform better. In cases where the Hopkins statistic of S.O. and DAN are similar (0.85 vs. 0.88), the difference in mutual information (0.57 vs. 0.70) provides justifications for the better performance of DAN.

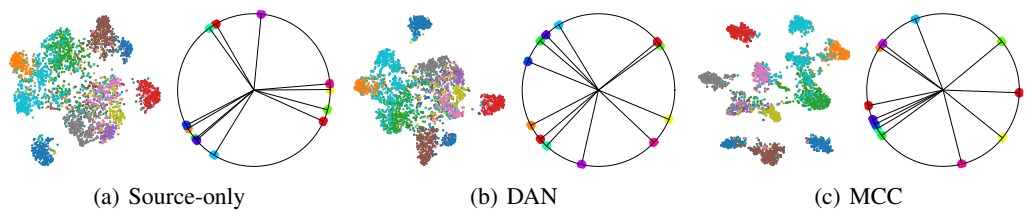

(a) Source-only

(b) DAN

(c) MCC

Figure 7: The t-SNE visualization and the uniformity of classifier on VisDA2017. From left to right, $\mathcal{H} : 0.85 \rightarrow 0.88 \rightarrow 0.93$, $\mathcal{M} : 0.57 \rightarrow 0.70 \rightarrow 0.89$, $\mathcal{U} : 0.1 \rightarrow 0.09 \rightarrow 0.05$. Greater $\mathcal{H}, \mathcal{M}$ indicates better clustering tendency, while less $\mathcal{U}$ indicates better uniformity.

# 6 CONCLUSION

UDA tackles the negative effect of data shift in machine learning. Previous UDA methods all rely on target-domain labels for model selection and tuning, which is not realistic in practice. This paper presents a solution to the evaluation challenge of UDA models in scenarios where target-domain labels are unavailable. To address this, we introduce a novel metric called the transfer score, which evaluates the uniformity of a classifier as well as the transferability and discriminability of features, represented by the Hopkins statistic and mutual information, respectively. Through extensive empirical analysis on four public UDA datasets, we demonstrate the efficacy of the TS in UDA model comparison, hyperparameter tuning, and checkpoint selection.

**Acknowledgements.** This work is supported by the NTU Presidential Postdoctoral Fellowship, "Adaptive Multimodal Learning for Robust Sensing and Recognition in Smart Cities" project fund, at Nanyang Technological University, Singapore. This research is jointly supported by the National Research Foundation, Singapore under its AI Singapore Programme (AISG Award No: AISG2-PhD-2021-08-008).

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

# A APPENDIX

## A.1 PROOF OF THEOREM 1

Suppose there are $K$ unit vectors $[w_1, w_2, ..., w_K]$ in a d-dimensional space $\mathbb{R}^d$, and the angles between any two unit vectors are equal, denote the angle in this case as $\theta_K$. Assume that $K \leq d + 1$. By expanding the sum of squares formula, we have

$$2 \sum_{i<j,i=1}^{K} \langle w_i, w_j \rangle = \left\| \sum_{i=1}^{K} w_i \right\|^2 - \sum_{i=1}^{K} \|w_i\|^2 \tag{7}$$

The first term on the right-hand side is greater than or equal to 0, and we have

$$2 \sum_{i<j,i=1}^{K} \langle w_i, w_j \rangle \geq - \sum_{i=1}^{K} \|w_i\|^2 = -K \tag{8}$$

When the angle between any two vectors is equal to K, the equation holds true, which can be simplified to

$$\langle w_i, w_j \rangle = -\frac{1}{K-1} \tag{9}$$

Thus,

$$\theta_K = arccos\left(-\frac{1}{K-1}\right). \tag{10}$$

Q.E.D.

## A.2 MORE RESULTS OF EMPIRICAL STUDIES

Here we put more empirical results of the proposed Transfer Score (TS) of unsupervised domain adaptation (UDA) on other datasets.

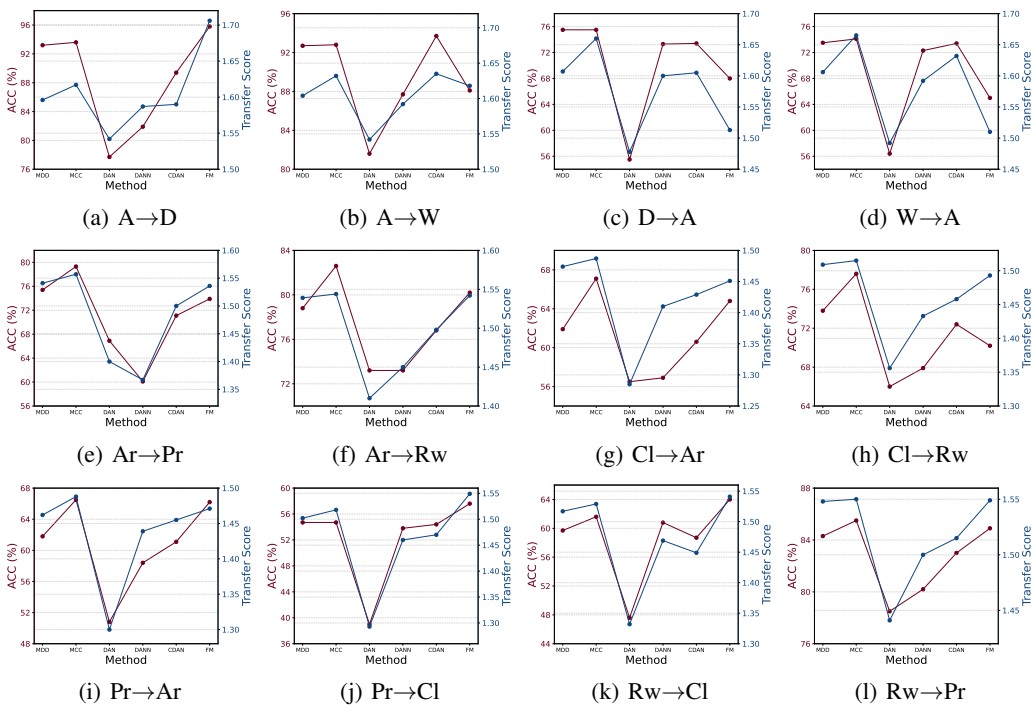

Figure 8: The relationship of accuracy and TS among different methods on 4 tasks of Office-31 and 8 tasks of Office-Home.

### A.2.1 Task 1 (UDA Method Comparison) on Office-31 and Office-Home

In this section, we add the experiments on all other tasks on Office-31 and Office-Home in Fig. 8. For Offce-31, we use all 4 tasks except the W→D and D→W as their transfer accuracy is over 99%. In most cases, the transfer score can help select the best-performing model across the candidates. Although in few cases, the difference in Transfer Scores does not perfectly align with the variations in accuracy, such as MCC (Jin et al., 2020) and MDD (Zhang et al., 2019), higher accuracy generally corresponds to better Transfer Scores. In A→D, D→A, and W→A, MCC achieves the highest Transfer Score and best performance.

Only in the A→W task, CDAN (Long et al., 2018) outperforms MCC by a margin of 0.9%, but its Transfer Score is relatively lower. However, the difference in TS between CDAN and MCC is very small, less than 0.3%, since they all perform well (>90%) in the A→W task, which makes it difficult for the metric to reflect such a small accuracy gap. Overall, TS still effectively reflects the quality of the models.

### A.2.2 Task 2 (Hyper-parameter Tuning) on VisDA-17 Dataset

Table 4: The relationship between TS and the target-domain accuracy using different hyper-parameters of MCC on VisDA-17. The models with better hyper-parameters are reflected by higher TS.

| Temperature | ACC (%) | Transfer Score |
|---|---|---|
| 0.1 | 55.5 | 1.179 |
| 1 | 71.7 | 1.801 |
| 3 | 76.5 | 1.819 |
| 9 | 56.1 | 1.706 |
| 27 | 51.0 | 1.014 |

We also investigate the impact of hyper-parameter in the MCC method. The temperature parameter in MCC is used for probability rescaling. In the original paper, the authors analyze the sensitivity of the hyper-parameter on the A→W task in the Office-31 dataset with access to the target-domain label and conclude that the optimal value for temperature is 3. From Tab. 4, we can draw similar conclusions, which are obtained from experiments conducted on a much larger dataset, VisDA-17. It is reasonable to believe that our proposed Transfer Score can help determine the approximate range of hyper-parameters without accessing the target labels.

### A.2.3 Task 3 (Epoch Selection) on VisDA-17 and DomainNet dataset

We further test the proposed method of selecting a good model checkpoint based on TS after UDA training on more tasks. We choose two datasets for our study: VisDA-17 and DomainNet, which include three tasks: Synthetic to Real, Clipart to Painting (c→p), and Real to Sketch (r→s). As shown in Fig. 9, for most methods, when the model overfits the target-domain, there is a certain decrease in accuracy. Our method can effectively capture the checkpoint before overfitting occurs. For SAFN (Xu et al., 2019) and MCC, our method is highly effective, providing an improvement of 5%-17% compared to selecting the last epoch, thus avoiding overtraining. For DAN (Long et al., 2017) and DANN (Ganin et al., 2016), the improvements are relatively smaller, ranging from 0.5% to 2%, possibly due to slower convergence or larger fluctuations in these models. Overall, our method can effectively prevent overfitting and identify a good checkpoint.

### A.3 Limitation

### A.3.1 Limitations in Task 1 (UDA Method Selection)

Table 5: Component analysis of the transfer score on Office-Home.

| Method | Ar→Cl $\mathcal{H}$ | $\mathcal{M}$ | $\mathcal{U}$ | Cl→Pr $\mathcal{H}$ | $\mathcal{M}$ | $\mathcal{U}$ | Pr→Rw $\mathcal{H}$ | $\mathcal{M}$ | $\mathcal{U}$ | Rw→Ar $\mathcal{H}$ | $\mathcal{M}$ | $\mathcal{U}$ |
|---|---|---|---|---|---|---|---|---|---|---|---|---|
| DANN | 0.838 | 0.667 | 0.066 | 0.846 | 0.678 | 0.069 | 0.855 | 0.696 | 0.066 | 0.829 | 0.690 | 0.066 |
| SAFN | 0.933 | 0.713 | 0.063 | 0.938 | 0.720 | 0.067 | 0.932 | 0.734 | 0.064 | 0.915 | 0.729 | 0.063 |
| MDD | 0.876 | 0.707 | 0.066 | 0.879 | 0.722 | 0.066 | 0.881 | 0.731 | 0.070 | 0.861 | 0.720 | 0.067 |

It is found that some methods cannot be measured accurately since they add part of the criterion that is directly relevant to the TS (e.g., mutual information) into the training procedure. For example, as

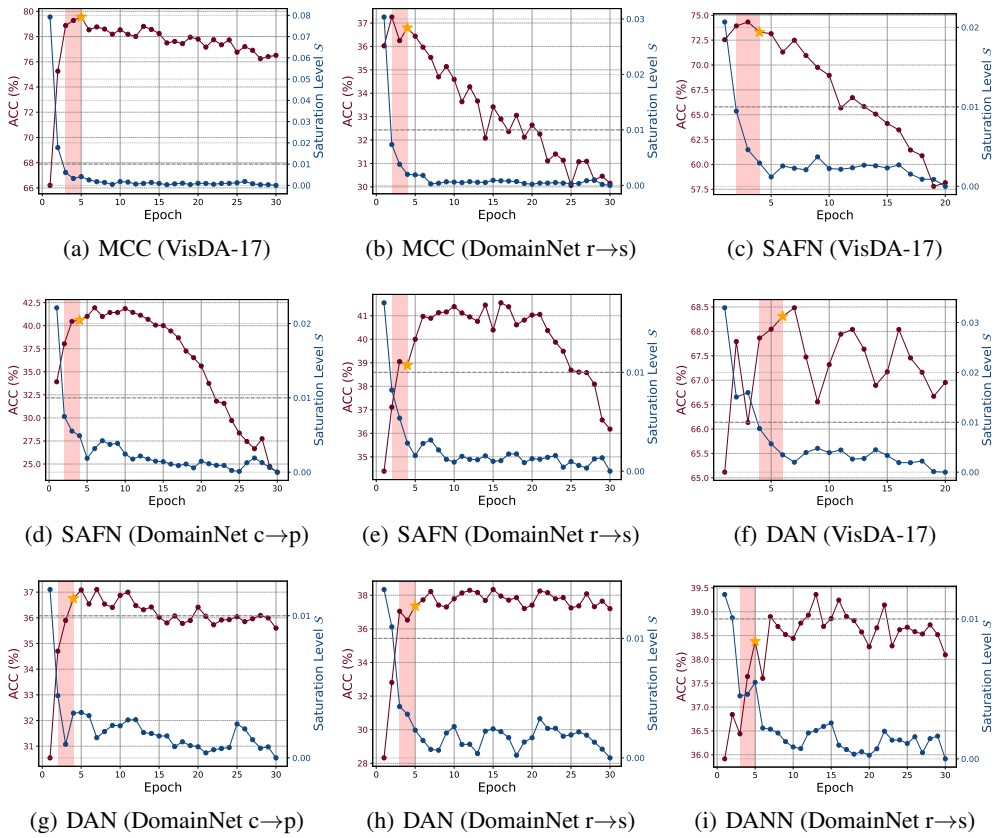

Figure 9: The choice of model checkpoint after UDA training for different methods.

shown in Tab. 5, we have observed that the Hopkins statistic value (Banerjee & Dave, 2004) $\mathcal{H}$ for SAFN becomes unusually high after few epochs of training, indicating a high degree of clustering in its features. This is because the regularization mechanism of SAFN not only encourages the enlargement of feature norms but also concentrates features around a fixed value. The concentration of feature norms further enhances the tendency for clustering. However, this does not lead to a high accuracy, because such clustering tendency is class-agnostic. Therefore, if we add some regularization to directly restrain one component of the TS, the TS might be less effective.

### A.3.2 LIMITATION IN TASK 3 (EPOCH SELECTION)

As shown in Fig. 10, the accuracy of MDD continuously increases without an early stopping point. In this case, our solution cannot find one of the best checkpoints but can provide a relatively cost-effective point. We also observe that the saturation level becomes smoother than those in Fig. 9 that encounter negative transfer. In this manner, a smoother convergence of saturation level might indicate a robust training curve without negative transfer, which will be explored in future work.

### A.4 IMPLEMENTATION DETAILS.

Table 6: Hyper-parameter settings for all the baseline methods.

| Hyper-parameter | DAN | DANN | CDAN | SAFN | MDD | MCC |
|---|---|---|---|---|---|---|
| LR | 0.003 | 0.01 | 0.01 | 0.001 | 0.004 | 0.005 |
| Trade-off | 1 | 1 | 1 | 0.1 | 1 | 1 |
| Others | - | - | - | $\Delta$r: 1 | Margin: 4 | Temperature: 3 |

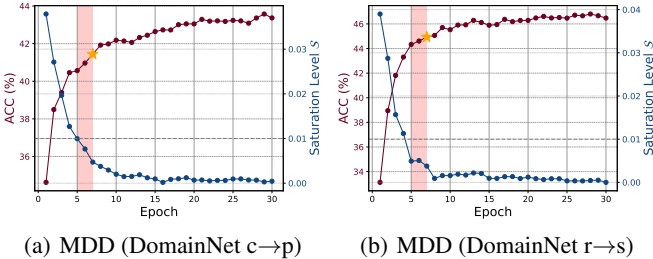

(a) MDD (DomainNet c→p)         (b) MDD (DomainNet r→s)

Figure 10: Failure cases of the epoch selection of the model (*i.e.*, task3).

We list all the hyper-parameters of our baseline methods in Tab. 6. We follow the original papers to set these hyper-parameters, except that we obtain some better hyper-parameters in terms of performances which are listed in the table. Note that the LR indicates the starting learning rate and the decay strategy is as same as those of the original papers.

## A.5  CORRELATION BETWEEN TRANSFER SCORE AND ACCURACY

To prove the better correlation with the accuracy, we conducted an experiment by comparing our metric with two advanced unsupervised validation metrics, C-Entropy and SND, on Office-31 and VisDA-17. The baseline models are DANN and MCC for Office-31 and VisDA-17, respectively. As shown in Fig. 11 and Fig. 12, our approach shows the best correlation coefficients. It is also observed that for VisDA-17, only our metric can reflect the overfitting issue while the other two metrics that keep increasing during training contradict with the decreasing accuracies.

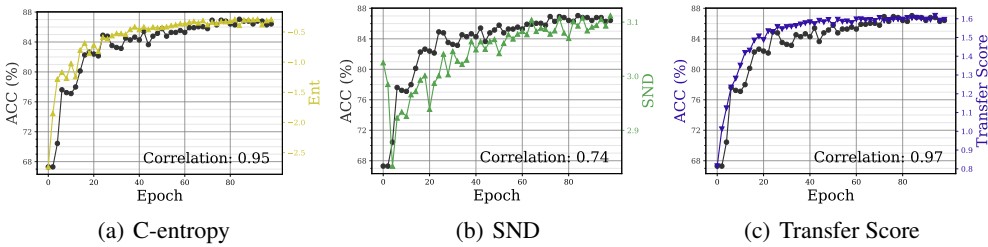

(a) C-entropy               (b) SND               (c) Transfer Score

Figure 11: The correlation between our method and existing transfer metrics. (DANN on Office-31 A→W)

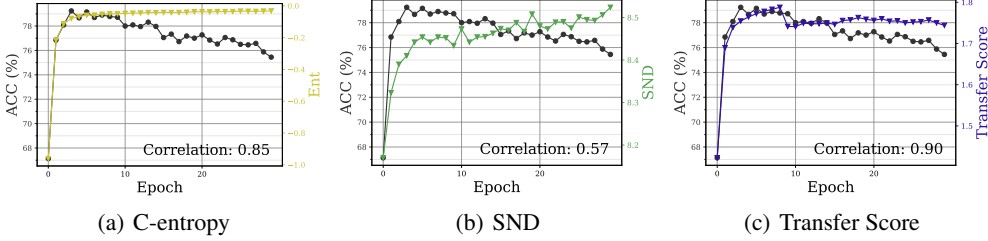

(a) C-entropy               (b) SND               (c) Transfer Score

Figure 12: The correlation between our method and existing transfer metrics. (MCC on VisDA-17)

## A.6  EPOCH SELECTION FOR MORE UDA METHODS

To further demonstrate the effectiveness of TS, we add four more UDA methods as baselines on VisDA-17. The results have been shown in Table 7. demonstrates that using transfer score can help select a better epoch, improving SHOT, CAN, CST, and AaD by 0.5%, 0.4%, 11.8%, and 2.5%, respectively.

Table 7: The comparison for epoch selection on VisDA-17.

| Method | Last | Ours | Imp. ↑ |
|--------|------|------|--------|
| SHOT | 77.5 | 78.0 | 0.5 |
| CAN | 86.4 | 86.8 | 0.4 |
| CST | 72.0 | 83.8 | 11.8 |
| AaD | 83.4 | 85.9 | 2.5 |

## A.7 EFFECTIVENESS OF TS ON SEMANTIC SEGMENTATION

We conducted an experiment on semantic segmentation. We use a classic cross-domain segmentation approach FDA (Yang & Soatto, 2020) as a baseline on an imbalanced dataset, GTA-5→Cityscapes. We train the FDA using the default hyper-parameters and use our TS to choose the best epoch of model. The mIoU results are shown in Table 8. It is shown that our metric can choose a better epoch of the model with an average mIoU of 39.7% while the last epoch of the model only achieves 37.7%. This demonstrates that our metric still works effectively on the imbalanced cross-domain semantic segmentation task.

Table 8: Evaluation on cross-domain segmentation (GTA-5→Cityscapes).

| Class | road | swalk | bding | wall | fence | pole | light | sign | vege | terrain | sky | person | rider | car | truck | bus | train | mtcyc | bicycle | Avg |
|-------|------|-------|-------|------|-------|------|-------|------|------|---------|-----|--------|-------|-----|-------|-----|-------|-------|---------|-----|
| Vanilla | 79.3 | 27.4 | 76.3 | 23.6 | 24.3 | 25.1 | 28.5 | 18.4 | 80.4 | 32.3 | 71.6 | 53.0 | 14.1 | 75.1 | 24.2 | 30.1 | 7.6 | 15.1 | 12.6 | 37.7 |
| Ours | 80.7 | 29.5 | 80.4 | 30.1 | 23.6 | 28.8 | 28.3 | 15 | 80.8 | 32.1 | 79.1 | 55.5 | 11.0 | 79.8 | 33.8 | 38.9 | 5.2 | 15.6 | 8.6 | **39.7** |

## A.8 DIFFERENCE WITH SND

We summarize the differences between our method and SND Saito et al. (2021) regarding the task, method and baseline UDA methods. Our work can achieve the model comparison while SND cannot. The transfer score measures the transferability from 3 perspectives while SND measures it via a single metric. In the empirical study, we prove the effectiveness of transfer score using more UDA baselines.

Table 9: The differences between our metric and SND.

| | Transfer Score (Ours) | SND |
|---|---|---|
| Task | Model comparison
Hyper-parameter tuning
Epoch selection | Epoch selection
Hyper-parameter tuning |
| Method | Transfer score is a three-fold metric, including uniformity of model weights, the mutual information of features, and the clustering tendency. | SND uses a single metric, the density of implicit local neighborhoods, which describes the clustering tendency. |
| Baseline UDA methods | 11 UDA methods:
Adversarial: CDAN, DANN, MDD
Moment matching: DAN, CAN
Reweighing-based: MCC
Self-training: FixMatch, CST, SHOT, AaD
Norm-based: SAFN | 4 methods:
Adversarial: CDAN
Reweighing-based: MCC
Self-training: NC, PL |

## A.9 USING TS AS A UDA REGULARIZER

add the experiment to explore the effectiveness of the three components of the transfer score. The experiments are conducted on Office-31 (A→W) and Office-Home (Ar→Cl) based on ResNet50. We add the three parts of transfer score as an independent regularizer on the target domain. As shown in the Table 10, it is shown that every component brings some improvement compared to the source-only

model. The total transfer score even brings significant improvement by 24.1% for A→W and 12.6% for A→W. The vanilla Hopkins statistic is calculated at the batch level so we think it can be enhanced further as a learning objective for UDA. We think this is an explorable direction in the future work.

Table 10: Evaluation using TS as a learning objective for UDA.

|  | A→W | Ar→Cl |
|---|---|---|
| Source-only | 68.4 | 34.9 |
| Uniformity | 75.1 | 41.8 |
| Hopkins statistic | 75.6 | 41.8 |
| Mutual information | 92 | 45.6 |
| Transfer Score (Total) | **92.5** | **47.6** |

### A.10   COMPARISON WITH SND AND C-ENTROPY ON TASK 1

Unsupervised model evaluation methods (e.g., SND Saito et al. (2021) and C-Entropy Morerio et al. (2018)) aim to select model parameters with better hyper-parameters for a specific UDA method. Though these works do not consider Task 1 in their original papers, we find that their scores can still be tested for Task 1. To this end, we conduct the experiments and calculate the SND and C-entropy of 6 UDA methods on three datasets including Office-31 (D→A), Office-Home (Ar→Pr), and VisDA-17. The results are shown in Fig. 13, where we mark the selected model for each metric with bold font. It is observed that the C-entropy succeeds in selecting the best model (MCC) on Office-31 but fails in Office-Home and VisDA-17, while SND fails all three datasets. In comparison, our proposed metric consistently selects the best UDA method for all three transfer tasks, significantly outperforming existing unsupervised validation methods.

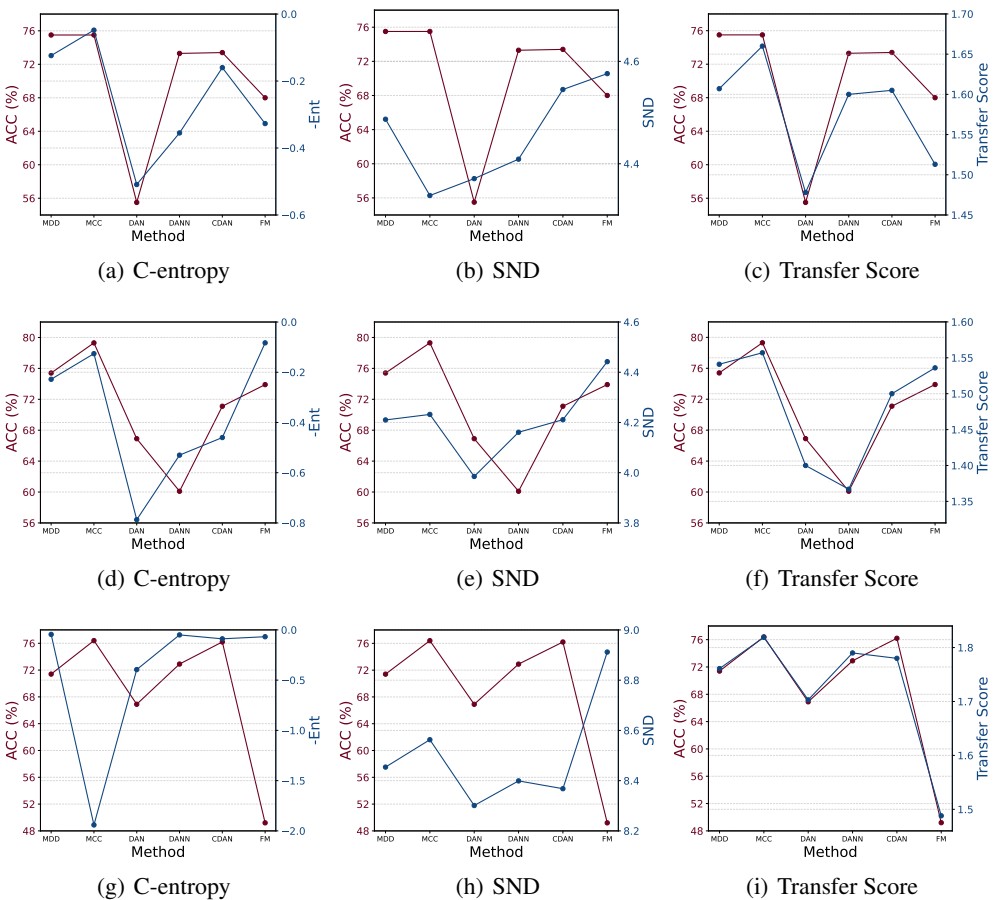

Figure 13: The comparison of UDA method comparison (Task 1). The three rows of figures are performed on Office-31 (D→A), Office-Home (Ar→Pr), and VisDA-17, respectively (top-down).

