# OpenReview forum: "Can We Evaluate Domain Adaptation Models Without Target-Domain Labels?"
_ICLR.cc/2024/Conference — ICLR 2024 poster_

### Official Review · Reviewer_VJy8 · 2023-10-30

**Soundness:** 3 good
**Presentation:** 3 good
**Contribution:** 2 fair
**Rating:** 6
**Confidence:** 4

**Summary:**

This paper studies the performance evaluation problem for unsupervised domain adaptation methods. Authors propose a evaluation metric called the Transfer Score. The propose metric consides both spatial uniformity from the model parameters and the transferability and discriminability calculated from the target samples. The metric can help users to select UDA methods, hyperparameters and checkpoints. Authors provide rich experiments to evalutate the effectiveness of the proposed metric.

**Strengths:**

1. The proposed Transfer Score is novel and the calculation process is clear.

2. The authors provide code in Supplementary Material, which makes the experiments convincing.

3. The structure of the paper is well organized.

**Weaknesses:**

1. Experimental results of baseline methods (e.g., DEV, SND) should be provided in the experiment section, there are only Transfer Score results of many dataset in that section.

2. Ablation study of Table 3 is weak, authors should provide the same one like Fig. 2 to evaluation the effectiveness of each term.

3. The representation of Equation 1 is incorrect, especially i*(i-1) and the use of i.

4. Authors should provide the complete results of all tasks from the different datasets * different UDA methods * baseline methods and proposed score, like Fig.3. The experimental results in the paper seem a bit sparse.

**Questions:**

1. What if the Transfer Score used as a optimization function for UDA task? I am curious about this.

2. Mutual information in Eq.4 is used as a term in the Transfer Score, if some methods (e.g. SHOT) minimize this term during adaptation, will it directly cause mutual information term to become invalid?

3. Because there is no source information used in Transfer Score, can authors provide the results on Source-Free Domain Adaptation methods?

---

> ### Author Response · Authors · 2023-11-20
> **Response to Reviewer VJy8 (1/3)**
>
> We sincerely thank the reviewer VJy8 for the insightful and constructive comments. We are glad that the reviewer acknowledges that the proposed metric is novel, the experiment is convincing, and the paper is well organized. In the rebuttal, we answer all the questions and add extensive experiments to improve our paper. We hope the responses can address the concerns. In addition, we have submitted a revised manuscript with an appendix where we mark all the suggested tables, figures, and analytics in magenta.
>
> **Q1: The experimental results of baseline methods should be provided in the experiment section. There are only Transfer Score results of many dataset.**
>
> **Answer**: Unsupervised model validation methods (DEV, SND) aim to select model parameters with better hyper-parameters (task 2&3) for the same UDA method,  e.g., SND, C-entropy, DEV, ATC that have been compared in the experiments. However, they cannot achieve Task 1 which performs comparison on different UDA methods. The reason is that existing metrics are not comparable as they do not have a specific range of value. As far as we know, the transfer score is the first work that performs unsupervised UDA method comparison.
>
> As suggested by the reviewer, task 2 should be compared with the SOTA method SND [1] (CVPR-21). To this end, we add two more experiments on DANN and SAFN using Office-31 (D$\to$A) and use transfer score and SND to select the hyper-parameters in an unsupervised manner. Note that the SND also studied the hyper-parameters of DANN so it is reasonable to make comparisons. As shown in the table below, the SND selects 62.4% while ours selects 67.8% on DANN. However, it is observed that SND fails to select a good hyper-parameter on the norm-based method SAFN. It contradicts the accuracy with a result of 31.3%, while ours still successfully selects 70.6%. Though the unsupervised metrics cannot fully align with the test accuracy, our transfer score can help select relatively high-performing hyper-parameters, outperforming SND significantly on hyper-parameter tuning. As for Task 3, the comparison with many baselines (DEV, SND, ATC, C-Entropy) has been discussed in Table 2 of the manuscript.
>
> | $\alpha$ (DANN) |   1   |   2   |     4     |   6   |     8     |   10  |   12  |   14  |   16  |
> |:---------------:|:-----:|:-----:|:---------:|:-----:|:---------:|:-----:|:-----:|:-----:|:-----:|
> |     Accuracy    |  71.6 |  71.8 |    67.8   |  67.0 |    62.4   |  57.9 |  64.0 |  64.8 |  42.4 |
> |       SND       | 4.447 | 4.448 |   4.486   | 4.470 | **4.537** | 4.370 | 4.420 | 4.321 | 4.306 |
> |       Ours      | 1.503 | 1.545 | **1.548** | 1.526 |   1.518   | 1.513 | 1.526 | 1.523 | 1.449 |
>
> | $\Delta r$ (SAFN) |   1   |     2     |   4   |   6   |   8   |   10  |   12  |   14  |     16    |
> |-------------------|:-----:|:---------:|:-----:|:-----:|:-----:|:-----:|:-----:|:-----:|:---------:|
> | Accuracy          |  72.2 |    70.6   |  66.3 |  57.6 |  50.7 |  46.8 |  40.2 |  33.5 |    31.3   |
> | SND               | 4.524 |   4.513   | 4.514 | 4.525 | 4.530 | 4.550 | 4.566 | 4.630 | **4.749** |
> | Ours              | 1.709 | **1.713** | 1.703 | 1.696 | 1.677 | 1.674 | 1.662 | 1.635 |   1.639   |
>
> We also supplement the relationship between the training curves and other baseline metrics. We supplement the experiment by comparing our metric with two advanced unsupervised validation metrics, C-Entropy and SND, on Office-31 and VisDA-17. The figures have been added to the revised manuscript (**Fig.11 and Fig.12**). For convenience, we also provide the table below to show the correlation coefficients between the metrics and the model accuracy as follows. For both datasets, our approach shows the best correlation coefficients. Fig.12 shows that for large-scale datasets like VisDA-17, only our metric can reflect the overfitting issue while the other two metrics that keep increasing during training contradict the decreasing accuracies.
>
> |                   | C-Entropy | SND  | Ours |
> |-------------------|-----------|------|------|
> | Office-31 A$\to$W | 0.95      | 0.74 | 0.97 |
> | VisDA-17          | 0.85      | 0.57 | 0.90 |

---

> ### Author Response · Authors · 2023-11-20
> **Response to Reviewer VJy8 (2/3)**
>
> **Q2: The authors should supplement the ablation study of Table 3.**
>
> **Answer**: We appreciate the suggestion and supplement two new UDA methods (CST [3] and AaD [4]) on a different dataset (VisDA-17) for the ablation study. As shown in the table below, the proposed transfer score selects the best model of 83.8% for CST and 85.9% for AaD. The results align with those of DomainNet. We have added the results to the revised manuscript in Table 3.
>
> | Uniformity   | Hopkins      | MI           | CST (VisDA-17) | AaD (VisDA-17) | MCC (DomainNet) |
> |--------------|--------------|--------------|----------------|----------------|-----------------|
> | $\checkmark$ |              |              | 81.8           | 84.9           | 35.8            |
> |              | $\checkmark$ |              | 82.2           | 84.9           | 35.8            |
> |              |              | $\checkmark$ | 82.7           | 84.9           | 35.9            |
> | $\checkmark$ | $\checkmark$ |              | **83.8**           | 85.3           | 36.0            |
> |              | $\checkmark$ | $\checkmark$ | 83.3           | 85.8           | 36.5            |
> | $\checkmark$ |              | $\checkmark$ | 83.3           | 85.6           | 36.5            |
> | $\checkmark$ | $\checkmark$ | $\checkmark$ | **83.8**       | **85.9**       | **37.3**        |
>
> **Q3: The Eq.1 is incorrect regarding the use of $i$.**
>
> **Answer**: We appreciate the careful check of the reviewer regarding the wrong formulas in Eq.(1). We have revised it in the manuscript.
>
> **Q4: Authors should provide the complete results of other tasks.**
>
> **Answer**: As suggested by the reviewer, we have added experiments on all other tasks of Office-31 and Office-Home. For Offce-31, we use all 4 tasks except the W$\to$D and D$\to$W as their transfer accuracy is over 99% which does not show the domain gap for UDA. For Office-Home, we supplement all 12 tasks. The results of Office-31 and Office-Home are shown as follows. In most cases, the transfer score can help select the best-performing model across the candidates on all tasks. Due to the page limit, these results have been summarized in Fig. 8 in the appendix of the revised manuscript.
>
> | Method |      A$\to$D     |      A$\to$W     |      D$\to$A     |      W$\to$A     |
> |:------:|:----------------:|:----------------:|:----------------:|:----------------:|
> |   DAN  |   77.7 / 1.542   |   81.6 / 1.542   |   55.5 / 1.478   |   56.4 / 1.492   |
> |  DANN  |   81.9 / 1.587   |   87.7 / 1.592   |   73.3 / 1.600   |   72.3 / 1.592   |
> |  CDAN  |   89.4 / 1.590   | **93.7 / 1.635** |   73.4 / 1.605   |   73.4 / 1.632   |
> |   MDD  |   93.2 / 1.596   |   92.7 / 1.604   |   75.5 / 1.607   |   73.5 / 1.606   |
> |   MCC  |   93.6 / 1.617   |   92.8 / 1.632   | **75.5 / 1.660** | **74.1 / 1.665** |
> |   FixMatch   | **95.8 / 1.706** |   88.1 / 1.618   |   68.0 / 1.531   |   65.0 / 1.510   |
>
> | Method   | A$\to$P          | A$\to$R          | C$\to$A          | C$\to$R          | P$\to$A          | P$\to$C          | R$\to$C          | R$\to$P          |
> |----------|------------------|------------------|------------------|------------------|------------------|------------------|------------------|------------------|
> | DAN      | 66.9 / 1.400     | 73.2 / 1.410     | 56.5 / 1.285     | 66 / 1.356       | 50.8 / 1.300     | 39.0 / 1.293     | 47.6 / 1.332     | 78.5 / 1.441     |
> | DANN     | 60.1 / 1.367     | 73.2 / 1.45      | 56.9 / 1.410     | 67.9 / 1.433     | 58.4 / 1.439     | 53.8 / 1.460     | 60.8 / 1.469     | 80.2 / 1.500     |
> | CDAN     | 71.1 / 1.500     | 76.8 / 1.498     | 60.6 / 1.429     | 72.4 / 1.458     | 61.1 / 1.455     | 54.4 / 1.470     | 58.7 / 1.449     | 83.0 / 1.515     |
> | MDD      | 75.4 / 1.541     | 78.8 / 1.539     | 61.9 / 1.474     | 73.8 / 1.509     | 61.8 / 1.462     | 54.7 / 1.502     | 59.7 / 1.517     | 84.3 / 1.548     |
> | MCC      | **79.3 / 1.557** | **82.6 / 1.544** | **67.1 / 1.487** | **77.6 / 1.515** | **66.5 / 1.488** | 54.7 / 1.518     | 61.6 / 1.529     | **85.5 / 1.550** |
> | FixMatch | 73.9 / 1.536     | 80.2 / 1.542     | 64.8 / 1.451     | 70.2 /1.493      | 66.2 / 1.471     | **57.6 / 1.549** | **64.0 / 1.541** | 84.9 / 1.549     |

---

> ### Author Response · Authors · 2023-11-20
> **Response to Reviewer VJy8 (3/3)**
>
> **Q5: What if the transfer score is used as a learning objective for UDA task? I am curious about this.**
>
> **Answer**: We appreciate the interesting suggestion and add the experiment to explore the effectiveness of the three components of the transfer score. The experiments are conducted on Office-31 (A$\to$W) and Office-Home (Ar$\to$Cl) based on ResNet50. We add the three parts of transfer score as an independent regularizer on the target domain. As shown in the table below, it is shown that every component brings some improvement compared to the source-only model. The total transfer score even brings significant improvement by 24.1% for A$\to$W and 12.6% for A$\to$W. The vanilla hopkins statistic is calculated at the batch level so it can be enhanced further as a learning objective for UDA. We think this is an explorable direction as an independent work in the future. The table and analytics have been added to the revised manuscript in Appendix A.10.
>
> |                        | A$\to$W | Ar$\to$Cl |
> |------------------------|:-------:|:---------:|
> | Source-only            |   68.4  |    34.9   |
> | Uniformity             |   75.1  |    41.8   |
> | Hopkins statistic      |   75.6  |    41.8   |
> | Mutual information     |    92   |    45.6   |
> | Transfer Score (Total) |   **92.5**  |    **47.6**   |
>
> **Q6+Q7: Some methods, e.g., SHOT, use the mutual information as an objective. Will it cause the proposed method become invalid? Can authors also provide some results on source-free domain adaptation (SFDA) methods (e.g., SHOT)?**
>
> **Answer**: Our method still works when Eq.4 (mutual information) is used as a learning objective. As we know, mutual information regularizer is used in SFDA methods (e.g., SHOT and AaD which uses a similar regularizer of Eq.4). Despite this, our proposed transfer score consists of three components, which is designed to avoid the single evaluation metric in previous unsupervised validation work (SND or C-Entropy). With three components, even though the mutual information is optimized by training, the other two metrics still work well. As suggested by the reviewer, we add experiments using two novel source-free UDA methods, SHOT (TPAMI’21) [2] and AaD (NeurIPS’22) [4], on Office-Home and VisDA-17. The results are shown as follows:
> - The first table shows the accuracy and the transfer score on Office-Home. The AaD achieves the best results on Office-Home with the highest transfer score.
> - The second table shows the improvement of epoch selection by transfer score on VisDA-17. Our method effectively selects a better epoch of the model for both methods, improving SHOT and AaD by 0.5% and 2.5%, respectively.
>
> | Method |   Ar$\to$Cl  |   Cl$\to$Pr  |   Pr$\to$Rw  |   Rw$\to$Ar  |
> |:------:|:------------:|:------------:|:------------:|:------------:|
> |  SHOT  | 56.7 / 1.532 | 74.0 / 1.522 | 82.1 / 1.570 | 72.3 / 1.544 |
> |   AaD  | 59.1 / 1.700 | 80.8 / 1.706 | 83.0 / 1.696 | 72.3 / 1.551 |
>
> | Method | Last Epoch | Ours | Imp. $\uparrow$ |
> |:------:|:----------:|:----:|:---------------:|
> |  SHOT  |    77.5    | 78.0 |       0.5       |
> |   AaD  |    83.4    | 85.9 |       2.5       |
>
> Therefore, the results show that the transfer score still works on source-free domain adaptation methods that leverage mutual information as a regularizer. We have added the results to Fig.8. As suggested by other reviewers, we also supplement the results on cross-domain semantic segmentation [1] and CAN [5] to the revised manuscript, which further demonstrates the effectiveness of transfer score on more tasks and totally 11 UDA baseline methods.
>
> **References**
>
> [1] Yang, Y. and Soatto, S., 2020. Fda: Fourier domain adaptation for semantic segmentation. In Proceedings of the IEEE/CVF conference on computer vision and pattern recognition (pp. 4085-4095).
>
> [2] Liang, J., Hu, D., Wang, Y., He, R. and Feng, J., 2021. Source data-absent unsupervised domain adaptation through hypothesis transfer and labeling transfer. IEEE Transactions on Pattern Analysis and Machine Intelligence, 44(11), pp.8602-8617.
>
> [3] Liu, H., Wang, J. and Long, M., 2021. Cycle self-training for domain adaptation. Advances in Neural Information Processing Systems, 34, pp.22968-22981.
>
> [4] Yang, S., Jui, S. and van de Weijer, J., 2022. Attracting and dispersing: A simple approach for source-free domain adaptation. Advances in Neural Information Processing Systems, 35, pp.5802-5815.
>
> [5] Kang, G., Jiang, L., Yang, Y. and Hauptmann, A.G., 2019. Contrastive adaptation network for unsupervised domain adaptation. In Proceedings of the IEEE/CVF conference on computer vision and pattern recognition (pp. 4893-4902).

---

> ### Author Response · Authors · 2023-11-22
> **Looking forward to your reply**
>
> Dear Reviewer VJy8,
>
> We have provided new experiments and discussions according to your valuable suggestions, which have been absorbed into the revised manuscript. We believe that the new manuscript is made to be stronger with your suggestions.
>
> As the rebuttal deadline is approaching, we look forward to your reply or any other suggestions. Thanks so much!
>
> Best Regards,
>
> Authors of Submission 7412

---

> ### Comment · Reviewer_VJy8 · 2023-11-22
> **Thanks to Author's reply**
>
> I appreciate the author's efforts in conducting additional experiments to address my queries. However, I still have some remaining questions:
>
> Q1: I think SND, Entropy and other UDA validation methods can be used for task1. They calculate a score for a specific task (UDA method+data+hyperparameters). The author's claim that these methods cannot be applied to task1 needs clarification.
> (The author did not provide citation information for SND and it should be a conference paper of ICCV21.)
>
> Q2: About using as an independent regularizer, I'm curious if using it as an evaluation method would be ineffective in this scenario.
>
> If the author can address my confusion, I would increase my score.

---

> ### Author Response · Authors · 2023-11-22
> **Thank you for the response and insightful questions**
>
> Dear Reviewer VJy8,
>
> We appreciate your response and conduct the experiments immediately. Here we provide the new results and analytics. We believe that these results let us have a better understanding of the proposed transfer score, and hope your concerns can be addressed accordingly.
>
> **Q1: I think SND, Entropy, and other UDA validation methods can be used for Task 1. They calculate a score for a specific task (UDA method+data+hyperparameters). The author's claim that these methods cannot be applied to Task 1 needs clarification. (The author did not provide citation information for SND and it should be a conference paper of ICCV21.)**
>
> We apologize for the unclear claim and our mistake. We agree with the reviewer that the scores of SND [1] and C-entropy [2] can be tested for Task 1 (i.e., UDA method comparison), though they do not consider Task 1 in their original papers. To this end, we conduct the experiments and calculate the SND and C-entropy of 6 UDA methods on three datasets including Office-31 (D$\to$A), Office-Home (Ar$\to$Pr), and VisDA-17. The results are shown in the tables below, where we mark the selected model for each metric with bold font. It is observed that the C-entropy succeeds in selecting the best model (MCC) on Office-31 but fails in Office-Home and VisDA-17, while SND fails all three datasets. In comparison, our proposed metric consistently selects the best UDA method for all three transfer tasks. *All the new results and analytics have been added to the appendix A.11 in Fig.13.*
>
> Regarding the citation, we apologize for the missing information and have included the correct reference for SND in our revised manuscript.
>
> Office-31 (D$\to$A)
>
> | Method | SND | C-Entropy | Ours | ACC |
> |:---:|:---:|:---:|:---:|:---:|
> | DAN | 4.371 | 0.509 | 1.478 | 55.5 |
> | DANN | 4.409 | 0.355 | 1.600 | 73.3 |
> | CDAN | 4.545 | 0.160 | 1.605 | 73.4 |
> | MDD | 4.487 | 0.124 | 1.607 | 75.5 |
> | MCC | 4.338 | **0.048** | **1.660** | **75.5** |
> | FixMatch | **4.576** | 0.327 | 1.531 | 68.0 |
>
> Office-Home (Ar$\to$Pr)
>
> | Method | SND | C-Entropy | Ours | ACC |
> |:---:|:---:|:---:|:---:|:---:|
> | DAN | 3.985 | 0.787 | 1.400 | 66.9 |
> | DANN | 4.161 | 0.530 | 1.367 | 60.1 |
> | CDAN | 4.211 | 0.460 | 1.500 | 71.1 |
> | MDD | 4.210 | 0.228 | 1.541 | 75.4 |
> | MCC | 4.232 | 0.126 | **1.557** | **79.3** |
> | FixMatch | **4.442** | **0.083** | 1.536 | 73.9 |
>
> VisDA-17
>
> | Method | SND | C-Entropy | Ours | ACC |
> |:---:|:---:|:---:|:---:|:---:|
> | DAN | 8.301 | 0.395 | 1.703 | 66.9 |
> | DANN | 8.399 | 0.050 | 1.790 | 72.9 |
> | CDAN | 8.368 | 0.089 | 1.780 | 76.2 |
> | MDD | 8.454 | **0.045** | 1.761 | 71.4 |
> | MCC | 8.563 | 1.941 | **1.819** | **76.4** |
> | FixMatch | **8.912** | 0.068 | 1.488 | 49.2 |
>
> **Q2: About using it as an independent regularizer, I'm curious if using it as an evaluation method would be ineffective in this scenario.**
>
> As suggested by the reviewer, we are happy to further explore this interesting problem. To this end, we perform the UDA model comparison on Office-31 (A$\to$W) and Office-Home (Ar$\to$Cl), and the model epoch selection on VisDA-17. Denote our new UDA method as TS-DA (Transfer Score for Domain Adaptation) in the table.
> - For the model comparison (in the 1st table), we show the accuracy and the transfer score. We notice that the highest transfer score still indicates the best model on two datasets. This raises a question: if we aim to optimize the TS, the TS of TS-DA should be the largest. Then we look back to the training procedure of TS-DA, and we find that the TS has converged during training. This inspires us that the current challenge of TS-DA lies in the optimization of TS. In future work, a new UDA method could be proposed to bring more improvement by revamping the optimization of TS to get higher TS for TS-DA.
> - For the epoch selection (in the 2nd table), our method still improves the vanilla TS-DA by 7.9% on VisDA-17, demonstrating the effectiveness of TS on TS-DA. The reason is that our epoch selection is based on the saturation level (variance) of the TS, which still works for TS-DA.
>
> | Method | A$\to$W | Ar$\to$Cl |
> |:---:|:---:|:---:|
> | TS-DA | 92.5 / 1.598 | 47.6 / 1.413 |
> | DAN | 81.6 / 1.542 | 44.2 / 1.321 |
> | DANN | 87.7 / 1.592 | 52.1 / 1.444 |
> | CDAN | **93.7 / 1.635** | 54.0 / 1.473 |
> | MDD | 92.7 / 1.604 | 55.9 / 1.518 |
> | MCC | 92.8 / 1.632 | **56.7 / 1.525** |
> | FM | 88.1 / 1.618 | 55.1 / 1.495 |
>
> | Method | Last | Ours | Imp. $\uparrow$ |
> |:---:|:---:|:---:|:---:|
> | TS-DA | 65.1 | 73.0 | 7.9 |
>
> **References**
>
> [1] Tune it the right way: Unsupervised validation of domain adaptation via soft neighborhood density. ICCV 2021.
>
> [2] Minimal-entropy correlation alignment for unsupervised deep domain adaptation. ICLR 2018.

---

> ### Author Response · Authors · 2023-11-23
> **Looking forward to your reply**
>
> Dear Reviewer VJy8,
>
> We have diligently addressed the remaining issues by conducting extensive experiments to support our responses. A revised manuscript has been submitted, accompanied by an appendix that delineates all revisions, highlighted in magenta color for clarity. We believe that the new manuscript is made to be stronger with your valuable suggestions.
>
> As the rebuttal deadline is approaching in several hours, we sincerely look forward to your reply. Thanks so much for your time and effort!
>
> Best Regards,
>
> Authors of Submission 7412

---

> ### Comment · Reviewer_VJy8 · 2023-11-23
> **Thanks for your reply**
>
> Thanks for author's reply. The experiment of task1 added by the author solved my previous concerns.
>
> I suggest the author update the description of task1 in the submission pdf and use UDA validation methods as some comparable baselines. It seems that task2 is similar.
>
>  I decide to improve my rating to 6.

---

> > ### Author Response · Authors · 2023-11-23
> > **Thanks for the appreciation**
> >
> > We sincerely appreciate the reply and will revise the manuscript pdf according to the advice. Thanks for your time and effort!

---

### Official Review · Reviewer_iLFW · 2023-10-31

**Soundness:** 3 good
**Presentation:** 3 good
**Contribution:** 3 good
**Rating:** 6
**Confidence:** 4

**Summary:**

This paper proposes a method named Transfer Score, which measures the classifier bias and feature discriminability, for the unsupervised validation problem.

**Strengths:**

- This idea is simple and easy to follow.

**Weaknesses:**

- The experiments are weak.
  - Task1 and Task2 are actually unsupervised model evaluation problems, all the unsupervised validation methods such as SND can be employed directly, but the authors do not compare their method with these methods.
  - In the experiments, only six UDA methods were evaluated. In task 2, there are only five candidate hyperparameters. The number is too small to illustrate the effectiveness of the proposed method.
  - The UDA datasets employed in the experiments are not comprehensive. On office-home, only four tasks were selected, and the situation was similar for DomainNet and Office. How do you select these tasks?

- The authors said, "TS is the **first** metric to perform simultaneous model comparison and selection without target-domain labels". As far as I know, the three tasks in experiments are actually unsupervised validation problems, and this is an existing research field. Besides, some out-of-distribution generalization prediction methods also solve the model selection problem without labels.
  > K-Means Clustering Based Feature Consistency Alignment for Label-Free Model Evaluation. \
  > Predicting Out-of-Distribution Error with Confidence Optimal Transport.\
  > Leveraging Unlabeled Data to Predict Out-of-Distribution Performance.\
  > ...

**Questions:**

See weakness.

---

> ### Author Response · Authors · 2023-11-20
> **Response to Reviewer iLFW (1/4)**
>
> We sincerely thank the reviewer iLFW for the detailed summary and constructive comments. We are glad that the reviewer acknowledges that the idea is simple and easy to follow. Here we answer all the questions and add extensive experiments for justification. We hope the responses can address the concerns. In addition, we have submitted a revised manuscript with an appendix where we mark all the suggested tables, figures, and analytics in magenta color.
>
> **Q1: Task1 and Task2 are actually unsupervised model evaluation problems, all the unsupervised validation methods such as SND can be employed directly, but the authors do not compare their methods.**
>
> **Answer**: We appreciate the valuable question. Unsupervised model evaluation methods (e.g., SND [1] and C-Entropy [7]) aim to select model parameters with better hyper-parameters for a specific UDA method. Though these works do not consider Task 1 in their original papers, as suggested by the reviewer, we find that their scores can still be tested for Task 1. To this end, we conduct the experiments and calculate the SND and C-entropy of 6 UDA methods on three datasets including Office-31 (D$\to$A), Office-Home (Ar$\to$Pr), and VisDA-17. The results are shown in the tables below, where we mark the selected model for each metric with bold font. It is observed that the C-entropy succeeds in selecting the best model (MCC) on Office-31 but fails in Office-Home and VisDA-17, while SND fails all three datasets. In comparison, our proposed metric consistently selects the best UDA method for all three transfer tasks, significantly outperforming existing unsupervised validation methods. All the new results and analytics have been added to the appendix A.11 in Fig.13.
>
> Office-31 (D$\to$A)
>
> | Method | SND | C-Entropy | Ours | ACC |
> |:---:|:---:|:---:|:---:|:---:|
> | DAN | 4.371 | 0.509 | 1.478 | 55.5 |
> | DANN | 4.409 | 0.355 | 1.600 | 73.3 |
> | CDAN | 4.545 | 0.160 | 1.605 | 73.4 |
> | MDD | 4.487 | 0.124 | 1.607 | 75.5 |
> | MCC | 4.338 | **0.048** | **1.660** | **75.5** |
> | FixMatch | **4.576** | 0.327 | 1.531 | 68.0 |
>
> Office-Home (Ar$\to$Pr)
>
> | Method | SND | C-Entropy | Ours | ACC |
> |:---:|:---:|:---:|:---:|:---:|
> | DAN | 3.985 | 0.787 | 1.400 | 66.9 |
> | DANN | 4.161 | 0.530 | 1.367 | 60.1 |
> | CDAN | 4.211 | 0.460 | 1.500 | 71.1 |
> | MDD | 4.210 | 0.228 | 1.541 | 75.4 |
> | MCC | 4.232 | 0.126 | **1.557** | **79.3** |
> | FixMatch | **4.442** | **0.083** | 1.536 | 73.9 |
>
> VisDA-17
>
> | Method | SND | C-Entropy | Ours | ACC |
> |:---:|:---:|:---:|:---:|:---:|
> | DAN | 8.301 | 0.395 | 1.703 | 66.9 |
> | DANN | 8.399 | 0.050 | 1.790 | 72.9 |
> | CDAN | 8.368 | 0.089 | 1.780 | 76.2 |
> | MDD | 8.454 | **0.045** | 1.761 | 71.4 |
> | MCC | 8.563 | 1.941 | **1.819** | **76.4** |
> | FixMatch | **8.912** | 0.068 | 1.488 | 49.2 |
>
> As suggested by the reviewer, Task 2 should be compared with the SOTA method SND [1] (CVPR-21). To this end, we add two more experiments on DANN and SAFN using Office-31 (D$\to$A) and use transfer score and SND to select the hyper-parameters in an unsupervised manner. Note that the SND also studied the hyper-parameters of DANN so it is reasonable to make comparisons. As shown in the table below, the SND selects 62.4% while ours selects 67.8% on DANN. However, it is observed that SND fails to select a good hyper-parameter on the norm-based method SAFN. It contradicts the accuracy with a result of 31.3%, while ours still successfully selects 70.6%. Though the unsupervised metrics cannot fully align with the test accuracy, our transfer score can help select relatively high-performing hyper-parameters, outperforming SND significantly on hyper-parameter tuning.
>
> | $\alpha$ (DANN) |   1   |   2   |     4     |   6   |     8     |   10  |   12  |   14  |   16  |
> |:---------------:|:-----:|:-----:|:---------:|:-----:|:---------:|:-----:|:-----:|:-----:|:-----:|
> | Accuracy    |  71.6 |  71.8 |    67.8   |  67.0 |    62.4   |  57.9 |  64.0 |  64.8 |  42.4 |
> |  SND | 4.447 | 4.448 |   4.486   | 4.470 | **4.537** | 4.370 | 4.420 | 4.321 | 4.306 |
> | Ours  | 1.503 | 1.545 | **1.548** | 1.526 |   1.518   | 1.513 | 1.526 | 1.523 | 1.449 |
>
> | $\Delta r$ (SAFN) |   1   |     2     |   4   |   6   |   8   |   10  |   12  |   14  |     16    |
> |-------------------|:-----:|:---------:|:-----:|:-----:|:-----:|:-----:|:-----:|:-----:|:---------:|
> | Accuracy   |  72.2 |    70.6   |  66.3 |  57.6 |  50.7 |  46.8 |  40.2 |  33.5 |    31.3   |
> | SND   | 4.524 |   4.513   | 4.514 | 4.525 | 4.530 | 4.550 | 4.566 | 4.630 | **4.749** |
> | Ours   | 1.709 | **1.713** | 1.703 | 1.696 | 1.677 | 1.674 | 1.662 | 1.635 |   1.639   |

---

> ### Author Response · Authors · 2023-11-20
> **Response to Reviewer iLFW (2/4)**
>
> We would like to summarize the difference between our metric and the existing unsupervised evaluation method (SND) in the table below. We have supplemented the experiments and the differences in the revised manuscript.
>
> |   | Ours   | SND    |
> |----------------------|-------------------------------------------------------------------------------------------------------------------------------------------------------------------------------------------------|-----------------------------------------------------------------------------------------------------------------|
> | Task | Model comparison Hyper-parameter tuning Epoch selection | Epoch selection Hyper-parameter tuning   |
> | Method  | Transfer score is a three-fold metric, including uniformity of model weights, the mutual information of features, and the clustering tendency.  | SND uses a single metric, the density of implicit local neighborhoods, which describes the clustering tendency. |
> | Baseline UDA methods | 11 UDA methods (including 4 new methods in the rebuttal): 1. Adversarial: CDAN, DANN, MDD 2. Moment matching: DAN, CAN 3. Reweighing-based: MCC 4. Self-training: FixMatch, CST, SHOT, AaD 5. Norm-based: SAFN | 4 UDA methods: 1. Adversarial: CDAN 2. Reweighing-based: MCC 3. Self-training: NC, PL   |
>
> **Q2: Only six UDA methods are evaluated and five candidate hyperparameters are provided. The number of these candidates should be increased.**
>
> **Answer**: We appreciate the suggestion and add extensive experiments to address this issue.
>
> Firstly, we supplement four new UDA methods including SHOT [2] (TPAMI-21), CST [3] (NeurIPS-21), AaD [4] (NeurIPS-22), and CAN [5] (CVPR-19). The experiments are conducted on Office-Home and VisDA-17 and we use their open-source codes as the backbones. The results have been shown as follows:
> - The first table shows the *accuracy* and *transfer score*. Our metric successfully selects the best-performing UDA method with the highest transfer score for these tasks.
> - The second table demonstrates that using transfer score can help select a better epoch on VisDA-17, improving SHOT, CAN, CST and AaD by 0.5%, 0.4%, 11.8%, and 2.5%, respectively. These comparisons have been summarized and updated in Fig.2 in the revised manuscript and the appendix.
>
> |  Method  |     Ar$\to$Cl    |     Cl$\to$Pr    |     Pr$\to$Rw    |     Rw$\to$Ar    |
> |:--------:|:----------------:|:----------------:|:----------------:|:----------------:|
> |    DAN   |   44.2 / 1.321   |   63.7 / 1.363   |   73.2 / 1.400   |   66.4 / 1.343   |
> |  CAN [5] |   50.1 / 1.390   |   70.8 / 1.381   |   76.3 / 1.422   |   68.5 / 1.412   |
> | SHOT [2] |   56.7 / 1.532   |   74.0 / 1.522   |   82.1 / 1.570   |   72.3 / 1.544   |
> |  CST [3] |   57.8 / 1.570   |   75.9 / 1.580   |    82.5/ 1.585   | **75.4 / 1.638** |
> |  AaD [4] | **59.1 / 1.700** | **80.8 / 1.706** | **83.0 / 1.696** |   72.3 / 1.551   |
>
> |  Method  | Last | Ours | Imp. $\uparrow$ |
> |:--------:|:----:|:----:|:---------------:|
> | SHOT [2] | 77.5 | 78.0 |       0.5       |
> |  CAN [5] | 86.4 | 86.8 |       0.4       |
> |  CST [3] | 72.0 | 83.8 |       11.8      |
> |  AaD [4] | 83.4 | 85.9 |       2.5       |
>
> In summary, we demonstrate that our method works effectively on four more UDA methods, two of which are source-free domain adaptation methods. Compared to the SOTA method SND, we have 11 UDA baselines to prove the effectiveness while SND only uses 4 UDA baselines.
>
> For the hyper-parameter candidate, we have enriched the candidate to [1, 2, 4, 6, 8, 10, 12, 14, 16] and the results have been shown in the first table in Q1. We demonstrate that our method can effectively select a better hyperparameter than SND.

---

> ### Author Response · Authors · 2023-11-20
> **Response to Reviewer iLFW (3/4)**
>
> **Q3: How do you select the transfer tasks on the datasets?**
>
> **Answer**: Following SND, we select some tasks that can contain all domains in Office-31 and Office-Home. The two tasks in DomainNet are selected as the evaluation of imbalanced long-tailed tasks. **As suggested by the reviewer, we realize that this is not comprehensive. Therefore we have added experiments on all other tasks of Office-31 and Office-Home.** For Offce-31, we use all 4 tasks except the W$\to$D and D$\to$W as their transfer accuracy is over 99% which does not show the domain gap for UDA. For Office-Home, we supplement all 12 tasks. The results of Office-31 and Office-Home are shown as follows (format: accuracy / transfer score). On Office-31, our method selects FixMatch, CDAN, and MCC for the best-performing models, which is very accurate. Our method also works effectively on all 12 tasks on Office-Home (whose table is displayed in the next paragraph). The results have been summarized in Fig. 8 in the appendix of the revised manuscript.
>
> | Method |      A$\to$D     |      A$\to$W     |      D$\to$A     |      W$\to$A     |
> |:------:|:----------------:|:----------------:|:----------------:|:----------------:|
> |   DAN  |   77.7 / 1.542   |   81.6 / 1.542   |   55.5 / 1.478   |   56.4 / 1.492   |
> |  DANN  |   81.9 / 1.587   |   87.7 / 1.592   |   73.3 / 1.600   |   72.3 / 1.592   |
> |  CDAN  |   89.4 / 1.590   | **93.7 / 1.635** |   73.4 / 1.605   |   73.4 / 1.632   |
> |   MDD  |   93.2 / 1.596   |   92.7 / 1.604   |   75.5 / 1.607   |   73.5 / 1.606   |
> |   MCC  |   93.6 / 1.617   |   92.8 / 1.632   | **75.5 / 1.660** | **74.1 / 1.665** |
> |   FixMatch   | **95.8 / 1.706** |   88.1 / 1.618   |   68.0 / 1.531   |   65.0 / 1.510   |
>
>
> Results on other 8 tasks on Office-Home:
>
> | Method   | A$\to$P          | A$\to$R          | C$\to$A          | C$\to$R          | P$\to$A          | P$\to$C          | R$\to$C          | R$\to$P          |
> |----------|------------------|------------------|------------------|------------------|------------------|------------------|------------------|------------------|
> | DAN      | 66.9 / 1.400     | 73.2 / 1.410     | 56.5 / 1.285     | 66 / 1.356       | 50.8 / 1.300     | 39.0 / 1.293     | 47.6 / 1.332     | 78.5 / 1.441     |
> | DANN     | 60.1 / 1.367     | 73.2 / 1.45      | 56.9 / 1.410     | 67.9 / 1.433     | 58.4 / 1.439     | 53.8 / 1.460     | 60.8 / 1.469     | 80.2 / 1.500     |
> | CDAN     | 71.1 / 1.500     | 76.8 / 1.498     | 60.6 / 1.429     | 72.4 / 1.458     | 61.1 / 1.455     | 54.4 / 1.470     | 58.7 / 1.449     | 83.0 / 1.515     |
> | MDD      | 75.4 / 1.541     | 78.8 / 1.539     | 61.9 / 1.474     | 73.8 / 1.509     | 61.8 / 1.462     | 54.7 / 1.502     | 59.7 / 1.517     | 84.3 / 1.548     |
> | MCC      | **79.3 / 1.557** | **82.6 / 1.544** | **67.1 / 1.487** | **77.6 / 1.515** | **66.5 / 1.488** | 54.7 / 1.518     | 61.6 / 1.529     | **85.5 / 1.550** |
> | FixMatch | 73.9 / 1.536     | 80.2 / 1.542     | 64.8 / 1.451     | 70.2 /1.493      | 66.2 / 1.471     | **57.6 / 1.549** | **64.0 / 1.541** | 84.9 / 1.549     |

---

> ### Author Response · Authors · 2023-11-22
> **Response to Reviewer iLFW (4/4)**
>
> **Q4:  The three tasks in experiments are actually unsupervised validation (UV) problems, and this is an existing research field. Besides, some out-of-distribution generalization prediction methods also solve the model selection problem without labels.**
>
> **Answer**:
> We apologize for not clearly illustrating the difference between existing unsupervised validation methods and ours. We would like to elaborate on the difference from three perspectives:
> 1. Existing UV methods (SND, C-entropy) aim at selecting hyper-parameters or epochs for the same UDA method, while our metric can conduct a UDA method comparison (Task 1) for the first time. We demonstrate in Q1 that other metrics cannot achieve the model comparison task. Therefore, we claim that our method is the first work to “perform simultaneous model comparison and selection without target-domain labels”.
> 2. For the Task 2&3, we demonstrate our method outperforms existing methods on more datasets using more categories of UDA methods. The reason is that our metric considers multiple aspects of the model while others use a single metric. For example, SND only uses the clustering characteristics and C-Entropy only uses the prediction uncertainty of samples.
> 3. We have already considered the UV methods in the out-of-distribution (OOD) research community. These methods are mostly based on sample-level uncertainty while ours rely on model-level and manifold-level metrics. The experiments have been made between ours and ATC (ICLR’22) [6] in Table 2. Our transfer score outperforms ATC on Office-Home and VisDA-17 based on two novel UDA baselines.
>
> **References**
>
> [1] Tune it the right way: Unsupervised validation of domain adaptation via soft neighborhood density. ICCV 2021.
>
> [2] Source data-absent unsupervised domain adaptation through hypothesis transfer and labeling transfer, T-PAMI 2021
>
> [3] Cycle self-training for domain adaptation, NeurIPS 2021
>
> [4] Yang S, Jui S, van de Weijer J. Attracting and dispersing: A simple approach for source-free domain adaptation. NeurIPS 2022
>
> [5] Kang G, Jiang L, Yang Y, et al. Contrastive adaptation network for unsupervised domain adaptation, CVPR 2019
>
> [6] Leveraging unlabeled data to predict out-of-distribution performance. ICLR 2022.
>
> [7] Minimal-entropy correlation alignment for unsupervised deep domain adaptation. ICLR 2018.

---

> ### Author Response · Authors · 2023-11-22
> **Looking forward to your reply**
>
> Dear Reviewer iLFW,
>
> We have diligently addressed all the issues by conducting extensive experiments to support our responses. A revised manuscript has been submitted, accompanied by an appendix that delineates all revisions, highlighted in magenta color for clarity. We believe that the new manuscript is made to be stronger with your valuable suggestions.
>
> As the rebuttal deadline is approaching, we sincerely look forward to your reply. Thanks so much for your time and effort!
>
> Best Regards,
>
> Authors of Submission 7412

---

> ### Comment · Reviewer_iLFW · 2023-11-23
> **Thanks for your response**
>
> I appreciate your experiments, which solve all my concerns and I will improve my rating.

---

> > ### Author Response · Authors · 2023-11-23
> > **Appreciation for the constructive suggestions**
> >
> > We sincerely appreciate your constructive comments which help us improve our manuscript. Thanks for your time and effort!

---

### Official Review · Reviewer_eAAs · 2023-11-02

**Soundness:** 2 fair
**Presentation:** 3 good
**Contribution:** 2 fair
**Rating:** 6
**Confidence:** 4

**Summary:**

Targeting at evaluating UDA models without target-domain labels, this paper proposes a new metric called the Transfer Score by assessing the spatial uniformity of the classifier via model parameters as well as the transferability and discriminability of deep representations. Three types of experiments are conducted for model evaluation.

**Strengths:**

+ The proposed metric is meaningful and intuitive.
+ The proposed metric seems useful in various evaluation settings.

**Weaknesses:**

- As the authors stated, “prevailing UDA methods relying on adversarial training and self-training could lead to model degeneration and negative transfer”, How can we prove this viewpoint? Can the proposed metric be applied to these types of methods for performance improvement in regular UDA settings?
- Although the proposed metric is intuitive, it is difficult to validate that it is definitely correct for evaluating a UDA model. In fact, various existing UDA approaches also adopt “Transfer Scores” for guiding the model learning process, such as [a]. A comparison between these metrics should be conducted.
- In figure 1, the correlation analysis is conducted only in one UDA task (Office31 A->W) and one UDA model (DANN), more comprehensive analysis and more comparison (with more metrics) should be conducted.
- In Definition 2, the design of the Score seems handcrafted. How can we ensure this formulation is optimal? Why use this design? The reasons should be analyzed.
- Some SOTA UDA approaches are not employed for experimental analysis [b-c].
- From Table 1, why are the improvements of the last four methods on the DomainNet (p->c) task so small?

[a] Active universal domain adaptation, ICCV 2021;
[b] Fixbi: Bridging domain spaces for unsupervised domain adaptation, CVPR 2021;
[c] Cycle self-training for domain adaptation, NeurIPS 2021.

**Questions:**

Please refer to the Weakness.

---

> ### Author Response · Authors · 2023-11-17
> **Response to Reviewer eAAs (1/2)**
>
> We sincerely appreciate the reviewer eAAs for the insightful and constructive comments. We are glad that the reviewer acknowledges that the proposed metric is meaningful and intuitive. Here we answer all the questions and add extensive experiments for justification. We hope the responses can address the concerns. In addition, we have submitted a revised manuscript with an appendix where we mark all the suggested tables, figures, and analytics in magenta color.
>
> **Q1: How can we prove that “prevailing UDA methods could lead to model degeneration and negative transfer”? Can the proposed metric be applied to them for performance improvement in regular UDA settings?**
>
> **Answer**: There are two probable situations when UDA methods lead to negative transfer:
> 1. inappropriate hyper-parameters: Since we do not have labeled target-domain samples for hyper-parameter tuning, the inappropriate hyper-parameters could lead to deteriorating convergence. As shown in Fig.3, for three UDA methods (DANN, SAFN, MDD), wrong hyper-parameters lead to even worse performances than the source-only model. This is usually caused by a greater weight of the transfer loss that dominates the training.
> 2. Overfitting to source domain: For large-scale UDA datasets, it is observed that the model is prone to overfitting to the source domain after a few epochs of training, leading to negative transfer. As shown in Fig.4, we see the overfitting happens on VisDA-17 and DomainNet. The SAFN even drops from 41% to 25% on DomainNet.
>
> The issue of negative transfer is also studied in some literature [7] that analyzes the reasons behind it. Our metric can effectively deal with the issues of negative transfer for regular UDA settings. In practice, as we do not have any target-domain labels, we usually use hyper-parameters and a fixed number of epochs sourced from the paper. However, the UDA method performs distinctly on different data, so the negative transfer may happen due to the aforementioned reasons. Our method is designed to work effectively in regular UDA settings. The task 2 and 3 in our work aim to overcome the issue by unsupervised validation on the target domain. The results in Table 1 and Fig.3 show the improvement of various UDA methods on different public datasets.
>
> **Q2: Some existing UDA methods also adopt a metric for guiding the model learning [1]. A comparison should be made.**
>
> **Answer**: We appreciate the suggestion and read the paper carefully. This paper proposes an active domain adaptation approach (CNTGE) based on a transferability metric. We would like to illustrate their differences regarding the tasks, method, and experiments.
>
> |  | Transfer Score | CNTGE |
> |---|---|---|
> | Task | Unsupervised validation of UDA methods | Active domain adaptation |
> | Method | Measure the uniformity of model weights, the mutual information of features, and the clustering tendency. | The metric proposed by CNTGE is used to select samples by the sample-level uncertainty of prediction. |
> | Significance | The transfer score is a generic metric to perform model comparison and selection for UDA methods. | The CNTGE is a novel active domain adaptation approach that achieves good results on benchmarks. |
>
> In the table, we see that the proposed metric is different from the sample-level transfer score in CNTGE. The transfer score proposed by CNTGE aims at selecting uncertain samples for active domain adaptation. It is a simple but effective metric for sample selection during UDA training, while our transfer score is a generic metric for unsupervised validation. Our metric measures the model-level transferability of the uniformity of model weights, the mutual information of features, and the clustering tendency. We have supplemented the contents in the manuscript.
>
> **Q3: Figure 1 is only conducted in one UDA task and model. More analysis should be conducted.**
>
> **Answer**: We appreciate this suggestion. The Fig.1 is originally regarded as an intuitive figure that shows the difference between existing metrics and our metric by a preliminary experiment. As suggested by the reviewer, we supplement the experiment by comparing our metric with two advanced unsupervised validation metrics, C-Entropy [8] and SND [9], on Office-31 (A$\to$W) and VisDA-17. Six figures have been added to the revised manuscript (*in the appendix, Fig.11 and Fig.12*). For convenience, we also provide the table below to show the **correlation coefficient** between the metrics and the model accuracy as follows. For both datasets, our approach shows the best correlation coefficients. Fig.12 shows that for large-scale datasets like VisDA-17, only our metric can reflect the overfitting issue while the other two metrics that keep increasing during training contradict the decreasing accuracies.
>
> |  | C-Entropy | SND | Ours |
> |---|---|---|---|
> | Office-31  | 0.95 | 0.74 | 0.97 |
> | VisDA-17 | 0.85 | 0.57 | 0.90 |

---

> ### Author Response · Authors · 2023-11-17
> **Response to Reviewer eAAs (2/2)**
>
> **Q4: In definition 2, how the authors design the score should be justified.**
>
> **Answer**: We appreciate the suggestions and apologize for the lack of justification. In definition 2, we aim to use a larger transfer score to indicate better transferability. To this end, as the greater uniformity indicates a larger bias and lower transferability, we use the negative uniformity. In contrast, we use the Hopkins statistic and the absolute value of mutual information as they are positively correlated to the transferability. The mutual information is especially normalized because it does not have a fixed range as the uniformity and Hopkins statistic does. We have included the justification of the design in the revised manuscript.
>
> **Q5: Some SOTA UDA approaches should be employed for experiments.**
>
> **Answer**: As suggested by the reviewer, we supplement four new UDA methods including SHOT [2] (TPAMI-21), CST [3] (NeurIPS-21), AAD [4] (NeurIPS-22), and CAN [5] (CVPR-19). The experiments are conducted on Office-Home and VisDA-17 and we use their open-source codes as the backbones. The results have been shown as follows:
> - The first table shows the **accuracy / transfer score** on Office-Home. Our metric successfully indicates the best-performing method with the highest transfer score for these tasks.
> - The second table demonstrates that using transfer score can help select a better epoch on VisDA-17, improving SHOT, CAN, CST, and AaD by 0.5%, 0.4%, 11.8%, and 2.5%, respectively. These comparisons have been summarized and updated in *Fig.2 in the revised manuscript and the appendix*.
>
> | Method | Ar$\to$Cl | Cl$\to$Pr | Pr$\to$Rw | Rw$\to$Ar |
> |:---:|:---:|:---:|:---:|:---:|
> | DAN | 44.2 / 1.321 | 63.7 / 1.363 | 73.2 / 1.400 | 66.4 / 1.343 |
> | CAN [5] | 50.1 / 1.390 | 70.8 / 1.381 | 76.3 / 1.422 | 68.5 / 1.412 |
> | SHOT [2] | 56.7 / 1.532 | 74.0 / 1.522 | 82.1 / 1.570 | 72.3 / 1.544 |
> | CST [3] | 57.8 / 1.570 | 75.9 / 1.580 | 82.5/ 1.585 | **75.4 / 1.638** |
> | AaD [4] | **59.1 / 1.700** | **80.8 / 1.706** | **83.0 / 1.696** | 72.3 / 1.551 |
>
> | Method | Last | Ours | Imp. $\uparrow$ |
> |:---:|:---:|:---:|:---:|
> | SHOT [2] | 77.5 | 78.0 | 0.5 |
> | CAN [5] | 86.4 | 86.8 | 0.4 |
> | CST [3] | 72.0 | 83.8 | 11.8 |
> | AaD [4] | 83.4 | 85.9 | 2.5 |
>
> In summary, we demonstrate that our method works effectively on four more UDA methods, two of which are even source-free domain adaptation methods. Note that we cannot reproduce Fixbi [6] since its open-source codes do not release its specially pre-trained DANN model parameters, which have been an issue raised and unsolved in Github repository. To ensure fair comparison we therefore decided not to use Fixbi.
>
> **Q6: In Table 1, why are the improvements of the last four methods on DomainNet (p$\to$c) so small?**
>
> **Answer**: In Table 1, the improvement of our method is caused by selecting a better epoch after training. Hence, the margin of improvement is limited by the range of varying accuracy during training using a specific UDA backbone. For example, on DomainNet (p->c), we find that DANN shown in Fig.4(e) has a decreasing accuracy from 40.2% to 34.5%, so when our metric selects the best epoch, the improvement is 5.7%. However, some methods like MDD and MCC show stable training curves w.r.t. the accuracy. Even though our method selects a better epoch, the improvement is marginal. The improvements are closely dependent on the specific training curve based on the datasets and the UDA model.
>
> **References**
>
> [1] Active universal domain adaptation, ICCV 2021
>
> [2] Source data-absent unsupervised domain adaptation through hypothesis transfer and labeling transfer, T-PAMI 2021
>
> [3] Cycle self-training for domain adaptation, NeurIPS 2021
>
> [4] Yang S, Jui S, van de Weijer J. Attracting and dispersing: A simple approach for source-free domain adaptation. NeurIPS 2022
>
> [5] Kang G, Jiang L, Yang Y, et al. Contrastive adaptation network for unsupervised domain adaptation, CVPR 2019
>
> [6] Fixbi: Bridging domain spaces for unsupervised domain adaptation, CVPR 2021
>
> [7] Characterizing and avoiding negative transfer. CVPR 2019
>
> [8] Minimal-entropy correlation alignment for unsupervised deep domain adaptation. arXiv preprint arXiv:1711.10288.
>
> [9] Tune it the right way: Unsupervised validation of domain adaptation via soft neighborhood density. CVPR 2021

---

> ### Author Response · Authors · 2023-11-20
> **Looking forward to your reply**
>
> Dear Reviewer eAAs,
>
> We have provided new experiments and discussions according to your valuable suggestions, which have been absorbed into the revised manuscript. We hope that the new manuscript is made to be stronger with your suggestions.
>
> As the rebuttal deadline is approaching, we look forward to your reply or any other suggestions. Thanks so much!
>
> Best Regards,
>
> Authors of Submission 7412

---

> > ### Comment · Reviewer_eAAs · 2023-11-21
> > **Thank you for your careful response**
> >
> > Thank you for your detailed answers to the other reviewers and me. The explanation and experiments are good for responding to most of my questions.
> > However, I still have some issues about: (1) Lack of comparison with different types of ``transfer score''. How can we ensure the score proposed by the authors is more effective? (2) Lack of theoretical analysis for the designed metric, which makes me think the proposed metric is still a heuristic one.
> > Note that I do not have strongly negative opinions about this paper, so I keep my original score and look forward to other reviewers' comments.

---

> ### Author Response · Authors · 2023-11-21
> **Thank you for the response**
>
> Dear reviewer,
>
> Thank you for your response. Regarding the new issues, we would like to address these issues by pointing out potential misunderstandings and misconceptions.
>
> > (1) Lack of comparison with different types of ``transfer score''. How can we ensure the score proposed by the authors is more effective?
>
> **We have already compared our method with four types of “transfer score”: (1) entropy-based score, C-entropy, (2) geometry-based score, SND, (3) out-of-distribution method, ATC, and (4) DEV for cross-validation.** We perform the comparison based on two UDA baselines and three UDA tasks on Office-31, Office-Home, and VisDA-17. Please refer to the results in Table 2 which demonstrates that our method outperforms these existing “transfer score” for model selection. Furthermore, as C-entropy [1] and SND [2] are also metric-based methods, we can intuitively draw the relationship between these metrics and the test accuracy, as shown in Fig.11 and Fig.12, which shows that our metric has the strongest correlation with the target-domain accuracy during training. **In summary, we argue that sufficient empirical studies have been made to show the superiority of the proposed metric against different types of "transfer score".**
>
> As suggested by your 1st review (Q2), to further demonstrate the effectiveness accordingly, we again add a new experiment using the sample-level “transfer score” (TS) [6] which has been discussed precisely in our 1st response. The sample-level metric cannot be directly used for model selection, so we use its average score across the whole target domain. The experiments are conducted on three challenging transfer tasks using DANN. Our metric outperforms the sample-level TS by 1.3%, 1.7% and 6.1% on VisDA-17, DomainNet (c$\to$p), and DomainNet (p$\to$c), respectively.
>
> | Metric | VisDA-17 | DomainNet (c$\to$p) | DomainNet (p$\to$c) |
> |---|---|---|---|
> | TS [6] | 72.5 | 36.2 | 34.1 |
> | Ours | **73.8** | **37.9** | **40.2** |
>
> > (2) Lack of theoretical analysis for the designed metric, which makes me think the proposed metric is still a heuristic one.
>
> We acknowledge that the current paper focuses more on empirical validation and practical utility in realistic domain adaptation applications, rather than an in-depth theoretical exposition of the metric. The basis of our metric stems from three perspectives: (1) the uniformity of the classifier weights, (2) the clustering tendency, and (3) the mutual information. We empirically show the strong capacity of our transfer score on model comparison and validation using many UDA approaches, outperforming existing unsupervised validation approaches. **We argue that it doesn't necessarily mean these heuristic metrics (including our metric, C-entropy [2], and SND [1]) lack novelty, but rather their theoretical foundations have not been thoroughly explored in the learning theory of domain adaptation [3][4]. We have also demonstrated that the metrics from the learning theory (i.e., the A-distance and maximum mean discrepancy [5]) cannot work for unsupervised model validation, as shown in Fig.1. In summary, we argue that the heuristic metrics have been commonly developed in unsupervised validation community, and should not be denoted as a shortcoming of our work.**
>
> *We sincerely appreciate the suggestions and positively supplement massive experiments to enhance the demonstration. We believe that our paper has been significantly improved after addressing the issues.*
>
> **References**
>
> [1] Minimal-entropy correlation alignment for unsupervised deep domain adaptation. ICLR 2018.
>
> [2] Tune it the right way: Unsupervised validation of domain adaptation via soft neighborhood density. CVPR 2021.
>
> [3] Analysis of representations for domain adaptation. NeurIPS 2006.
>
> [4] Redko, I., Morvant, E., Habrard, A., Sebban, M., & Bennani, Y. (2019). Advances in domain adaptation theory. Elsevier.
>
> [5] A kernel two-sample test. The Journal of Machine Learning Research, 13(1), 723-773.
>
> [6] Active universal domain adaptation, ICCV 2021

---

> > ### Comment · Reviewer_eAAs · 2023-11-22
> > **Thank you for your response**
> >
> > Thank you for your response. I have improved my rating to 6.

---

> > > ### Author Response · Authors · 2023-11-22
> > > **Appreciation for the constructive comments**
> > >
> > > We sincerely appreciate your constructive comments and prompt response which help us improve our paper.

---

### Official Review · Reviewer_hZ7S · 2023-11-07

**Soundness:** 3 good
**Presentation:** 3 good
**Contribution:** 3 good
**Rating:** 6
**Confidence:** 5

**Summary:**

To evaluate the performance of the Unsupervised Domain Adaptation (UDA) model, this paper proposes a novel metric called “Transfer Score”. Based on this metric, we achieve three novel objectives without target-domain labels: (1) selecting the best UDA method (2) optimizing the hyperparameter of the UDA model (3) selecting the best checkpoint.

**Strengths:**

1.	Evaluate UDA models in an unsupervised manner is very important, but there is a lack of relevant research in the current community
2.	Experiments demonstrate the effectiveness of “Transfer Score” in method selection, hyperparameter tuning, and checkpoint selection.
3.	The paper is well-written and easy to understand.

**Weaknesses:**

1.	“Transfer Score” uses clustering and class balance as the measurement criteria. However, the assumption of clustering and class balance usually may not always hold. Acutally, in many real world tasks, class imbanlance often exists. For example, in cross-domain semantic segmentation, a severe category imbalance is often present [1], which limits the application of this metric.
2.	Some methods[2,3] directly adopt Eq. 1 and Eq. 4 as optimization objectives. For these methods, the transfer score may be invalid.
3.	The author needs to verify the effectiveness of the “Transfer Score” on more advanced UDA methods[3,4].

[1] Liu Y, Deng J, Tao J, et al. Undoing the damage of label shift for cross-domain semantic segmentation[C]//Proceedings of the IEEE/CVF Conference on Computer Vision and Pattern Recognition. 2022: 7042-7052.
[2] Liang J, Hu D, Wang Y, et al. Source data-absent unsupervised domain adaptation through hypothesis transfer and labeling transfer[J]. IEEE Transactions on Pattern Analysis and Machine Intelligence, 2021, 44(11): 8602-8617.
[3] Yang S, Jui S, van de Weijer J. Attracting and dispersing: A simple approach for source-free domain adaptation[J]. Advances in Neural Information Processing Systems, 2022, 35: 5802-5815.
[4] Kang G, Jiang L, Yang Y, et al. Contrastive adaptation network for unsupervised domain adaptation[C]//Proceedings of the IEEE/CVF conference on computer vision and pattern recognition. 2019: 4893-4902.

**Questions:**

I would like to see authors' response regarding the weaknesses.

---

> ### Author Response · Authors · 2023-11-17
> **Response to Reviewer hZ7S (1/2)**
>
> We sincerely thank the reviewer hZ7S for the insightful and constructive comments. We are glad that the reviewer acknowledges that the problem is important and the method is effective on three tasks. Here we answer all the questions and add extensive experiments for justification. We hope the responses can address the concerns. In addition, we have submitted a revised manuscript with an appendix where we mark all the suggested tables, figures, and analytics in magenta color.
>
> **Q1: “Transfer Score” uses clustering and class balance as the measurement criteria. Does the transfer score apply to the imbalanced UDA task, such as the cross-domain classification and semantic segmentation?**
>
> **Answer**: Yes, the proposed transfer score works effectively on imbalanced UDA tasks. As illustrated in Section 5.5, to validate the effectiveness of handling imbalanced datasets, we have conducted an experiment on DomainNet (two transfer tasks) where severe imbalanced class distribution exists as shown in Fig.5(a). Our method still shows good results as shown in Table 1, Fig.5(b) and Fig.6. According to the reviewer’s suggestion, to better evaluate the imbalanced situation, we add a new experiment on semantic segmentation. We use a classic cross-domain segmentation approach FDA [1] as a baseline on GTA-5 -> Cityscapes. The mIoU results are shown below. It is shown that our metric can choose a better epoch of the model with an average mIoU of 39.7% while the last epoch of the model only achieves 37.7%. This demonstrates that our metric still works effectively on the imbalanced cross-domain semantic segmentation task.
>
> | Class | road | sidewalk | building | wall | fence | pole | light | sign | vegetation | terrain | sky | person | rider | car | truck | bus | train | motocycle | bicycle | Avg |
> |:---:|:---:|:---:|:---:|:---:|:---:|:---:|:---:|:---:|:---:|:---:|:---:|:---:|:---:|:---:|:---:|:---:|:---:|:---:|:---:|:---:|
> | Vanilla | 79.3 | 27.4 | 76.3 | 23.6 | 24.3 | 25.1 | 28.5 | 18.4 | 80.4 | 32.3 | 71.6 | 53.0 | 14.1 | 75.1 | 24.2 | 30.1 | 7.6 | 15.1 | 12.6 | 37.7 |
> | Ours | 80.7 | 29.5 | 80.4 | 30.1 | 23.6 | 28.8 | 28.3 | 15 | 80.8 | 32.1 | 79.1 | 55.5 | 11.0 | 79.8 | 33.8 | 38.9 | 5.2 | 15.6 | 8.6 | **39.7** |
>
> Intuitively, the clustering metric (Hopkins statistic) measures the clustering tendency, which is irrelevant to class balance. The uniformity measures the classifier bias, i.e., how uniformly the feature space is divided by a classifier. It is directly calculated on model weights and does not measure the class balance. For UDA, the better uniformity of a classifier indicates better generalization ability. For example, if a classifier overfits the source domain, it should fail to perform well on the target domain with a different class distribution, and such a classifier has a low level of uniformity.
>
> **Q2: Does the transfer score work in some methods (SHOT, AAD) that directly use Eq.1 and Eq.4 as learning objectives?**
>
> **Answer**: Yes. Our method still works when Eq.4 (mutual information) is used as a learning objective. As mentioned by the reviewer, mutual information regularizer is very commonly used in source-free UDA methods (e.g., SHOT and AaD which uses a similar regularizer of Eq.4). Whereas, our proposed transfer score consists of three components, which is designed to avoid the single evaluation metric in previous unsupervised validation work. With three components, even though the mutual information is optimized by training, the other two metrics still work well. To prove it, we supplement the experiments using SHOT [2] and AaD [4] on Office-Home and VisDA-17. The results are shown as follows:
> - The first table shows the *accuracy* and *transfer score* on Office-Home. The AaD achieves the best results on Office-Home with the highest transfer score.
> - The second table shows the improvement of epoch selection by transfer score on VisDA-17. Our method effectively selects a better epoch of the model for both methods, improving SHOT and AaD by 0.5% and 2.5%, respectively.
> Therefore, the results show that the transfer score still works when Eq.4 is used as a learning objective. Eq.1 is proposed by us in this paper, and we have not found any methods using it as a learning objective.
>
> | Method | Ar$\to$Cl | Cl$\to$Pr | Pr$\to$Rw | Rw$\to$Ar |
> |:---:|:---:|:---:|:---:|:---:|
> | SHOT [2] | 56.7 / 1.532 | 74.0 / 1.522 | 82.1 / 1.570 | 72.3 / 1.544 |
> | AaD [4] | 59.1 / 1.700 | 80.8 / 1.706 | 83.0 / 1.696 | 72.3 / 1.551 |
>
> | Method | Last | Ours | Imp. $\uparrow$ |
> |:---:|:---:|:---:|:---:|
> | SHOT [2] | 77.5 | 78.0 | 0.5 |
> | AaD [4] | 83.4 | 85.9 | 2.5 |

---

> ### Author Response · Authors · 2023-11-17
> **Response to Reviewer hZ7S (2/2)**
>
> **Q3: The authors should verify the effectiveness of the “Transfer Score” on two advanced UDA methods (AAD, CAN).**
>
> As suggested by the reviewer, we have added the experiments on AaD [4] and CAN [5]. We also included an advanced UDA method CST [3] advised by another reviewer. The experiments are conducted on Office-Home and VisDA-17 and we use their open-source codes as the backbones. The results are shown as follows. The first table shows that the transfer score successfully indicates the best-performing method for these tasks. The second table demonstrates that using transfer score can help select a better epoch, improving CAN, CST, and AaD by 0.4%, 11.8%, and 2.5%, respectively. These comparisons have been updated in the revised manuscript in Fig.2 and the appendix.
>
> | Method | Ar$\to$Cl | Cl$\to$Pr | Pr$\to$Rw | Rw$\to$Ar |
> |:---:|:---:|:---:|:---:|:---:|
> | DAN | 44.2 / 1.321 | 63.7 / 1.363 | 73.2 / 1.400 | 66.4 / 1.343 |
> | CAN [5] | 50.1 / 1.390 | 70.8 / 1.381 | 76.3 / 1.422 | 68.5 / 1.412 |
> | CST [3] | 57.8 / 1.570 | 75.9 / 1.580 | 82.5 / 1.585 | **75.4 / 1.638** |
> | AaD [4] | **59.1 / 1.700** | **80.8 / 1.706** | **83.0 / 1.696** | 72.3 / 1.551 |
>
> | Method | Last | Ours | Imp. $\uparrow$ |
> |:---:|:---:|:---:|:---:|
> | CAN [5] | 86.4 | 86.8 | 0.4 |
> | CST [3] | 72.0 | 83.8 | 11.8 |
> | AaD [4] | 83.4 | 85.9 | 2.5 |
>
> **References**
>
> [1] Yang, Y. and Soatto, S., 2020. Fda: Fourier domain adaptation for semantic segmentation. In Proceedings of the IEEE/CVF conference on computer vision and pattern recognition (pp. 4085-4095).
>
> [2] Liang, J., Hu, D., Wang, Y., He, R. and Feng, J., 2021. Source data-absent unsupervised domain adaptation through hypothesis transfer and labeling transfer. IEEE Transactions on Pattern Analysis and Machine Intelligence, 44(11), pp.8602-8617.
>
> [3] Liu, H., Wang, J. and Long, M., 2021. Cycle self-training for domain adaptation. Advances in Neural Information Processing Systems, 34, pp.22968-22981.
>
> [4] Yang, S., Jui, S. and van de Weijer, J., 2022. Attracting and dispersing: A simple approach for source-free domain adaptation. Advances in Neural Information Processing Systems, 35, pp.5802-5815.
>
> [5] Kang, G., Jiang, L., Yang, Y. and Hauptmann, A.G., 2019. Contrastive adaptation network for unsupervised domain adaptation. In Proceedings of the IEEE/CVF conference on computer vision and pattern recognition (pp. 4893-4902).

---

> ### Author Response · Authors · 2023-11-20
> **Looking forward to your reply**
>
> Dear Reviewer hZ7S,
>
> We have provided new experiments and discussions according to your valuable suggestions, which have been absorbed into the revised manuscript. We hope that the new manuscript is made to be stronger with your suggestions.
>
> As the rebuttal deadline is approaching, we look forward to your reply or any other suggestions. Thanks so much!
>
> Best Regards,
>
> Authors of Submission 7412

---

### Author Response · Authors · 2023-11-22
**Summary of the rebuttal and the major changes of revised manuscript**

Dear reviewers,

We would like to express our heartfelt gratitude for your invaluable time, expertise, and meticulous attention in reviewing our manuscript. The insightful comments and constructive feedback have immensely enriched the quality and rigor of our work.

We appreciate that the reviewers acknowledge the advantages of our work: **“the problem is important and the method is effective”** (reviewer hZ7S), **“the proposed metric is meaningful and intuitive”** (reviewer eAAs), **“the idea is simple and easy to follow”** (reviewer iLFW), **“the method is novel and the code is public, which makes the experiments more convincing”** (reviewer VJy8).

On the other hand, we have diligently addressed all the issues by conducting extensive experiments to support our responses. A revised manuscript has been submitted, accompanied by an appendix that delineates all revisions, highlighted in magenta color for clarity. Owing to space constraints, selected experiments have been incorporated into the main manuscript while supplementary experiments have been included in the appendix. Allow me to summarize the significant alterations made in both the rebuttal and the revised manuscript:

1. **Expanded Baselines**: Included four additional UDA methods (SHOT, CST, AaD, CAN) as baselines to demonstrate the effectiveness of the proposed method in source-free domain adaptation.
2. **Effectiveness Demonstration on Cross-domain Semantic Segmentation**: Introduced a new experiment focusing on cross-domain semantic segmentation (specifically GTA5$\to$Cityscapes) to highlight the effectiveness of the approach in this task.
3. **Comprehensive Results on All Tasks**: Incorporated comprehensive results from all tasks within Office-31 and Office-Home datasets, offering a broader perspective on the performance of the proposed method across various tasks.
4. **Ablation Study Enrichment**: Expanded the ablation study by including two more UDA methods and tasks, providing a more detailed analysis of the proposed method's performance under varied conditions.
5. **Hyper-parameter Candidates Enrichment**: Enhanced the Task 2 (hyper-parameter tuning) experiment by testing and exploring a wider range of hyper-parameter candidates, potentially offering deeper insights into the method's sensitivity to parameter settings.
6. **Intuitive Visualization**: Provided visualizations demonstrating the correlation between other comparative metrics and test accuracy in Fig.11 and Fig.12, offering additional insights into the evaluation metrics and their relationship to the proposed method's performance.
7. **Writing Revision and Explanation**: Revised formula errors and provided a clearer explanation and intuition behind the design of the proposed metric or method.
8. **Comparison with Related Methods**: Explored and highlighted the differences between the proposed metric and other existing metrics by experiments on model comparison and selection, emphasizing the uniqueness and advantages of the introduced metric.
9. **Working As UDA Regularizer**: Introduced an experiment showcasing that using TS as a learning objective resulted in significant improvements across UDA tasks, highlighting the method's efficacy as an objective function.

&nbsp;

Best Regards,

Authors of Submission 7412

---

### Meta-Review · Area_Chair_auH5 · 2023-12-06

**Metareview:**

The paper proposes Transfer Score, a novel metric for evaluating Unsupervised Domain Adaptation (UDA) models without target-domain labels. It considers spatial uniformity, classifier bias, and feature discriminability in deep representations. The metric aids in selecting UDA methods, hyperparameters, and checkpoints, demonstrated through comprehensive experiments. The author has done a well rebuttal and solved the previous concerns. There have three reviewers raised their final score. Hence, considering the attitude of all reviewers, I recommend acceptance.

**Justification For Why Not Higher Score:**

This work provides a novel metric for evaluating Unsupervised Domain Adaptation (UDA) models, it has some insights. However, it is not enough to be selected as a spotlight.

**Justification For Why Not Lower Score:**

The paper is well-organized and easy to follow. The author has done a well rebuttal and solved the previous concerns.

---

### Decision · Program_Chairs · 2024-01-16

Accept (poster)